



# Retrieval of atmospheric $CO_2$ vertical profiles from ground-based near-infrared spectra

Sébastien Roche[1], Kimberly Strong[1], Debra Wunch[1], Joseph Mendonca[2], Colm Sweeney[4], Bianca Baier[3,4], Sébastien C. Biraud[5], Joshua L. Laughner[6], Geoffrey C. Toon[7], and Brian J. Connor[8]

[1]Department of Physics, University of Toronto, Toronto, ON, Canada
[2]Climate Research Division, Environment and Climate Change Canada, Toronto, ON, Canada
[3]Cooperative Institute for Research in Environmental Sciences, University of Colorado, Boulder, CO, USA
[4]NOAA Global Monitoring Laboratory, Boulder, CO, USA
[5]Lawrence Berkeley National Laboratory, Berkeley, CA, USA
[6]California Institute of Technology, Pasadena, CA, USA
[7]Jet Propulsion Laboratory, California Institute of Technology, Pasadena, CA, USA
[8]BC Consulting Limited, Martinborough, New Zealand

*Correspondence to:* Sébastien Roche (sebastien.roche@mail.utoronto.ca)

**Abstract.** We evaluate vertical profile retrievals of $CO_2$ from 0.02 cm$^{-1}$ resolution ground-based near-infrared solar absorption spectra with the GFIT2 algorithm, using improved spectroscopic linelists and line shapes. With these improvements, $CO_2$ profiles were obtained from sequential retrievals in five spectral windows with different vertical sensitivities. A sensitivity study using synthetic spectra shows that the leading source of uncertainty in the retrieved $CO_2$ profiles is the error in the a priori temperature profile, even with 3-hourly reanalysis a priori profiles. A 2°C error in the temperature profile in the lower troposphere between 0.6 and 0.85 atm causes deviations in the retrieved $CO_2$ profiles that are larger than the typical vertical variations of $CO_2$. To distinguish the effect of errors in the a priori meteorology and trace gas concentration profiles from those in the instrument alignment and spectroscopic parameters, we retrieve $CO_2$ profiles from atmospheric spectra while using an a priori built from coincident AirCore, radiosonde, and surface in situ measurements at the Lamont, Oklahoma (USA) Total Carbon Column Observing Network station. In those cases, the deviations in retrieved $CO_2$ profiles are also larger than typical vertical variations of $CO_2$, suggesting that remaining errors in the forward model limit the accuracy of the retrieved profiles. Implementing a temperature retrieval or correction, and quantifying and modeling an imperfect instrument alignment, are critical to improve $CO_2$ profile retrievals. Without significant advances in modeling imperfect instrument alignment, and improvements in the accuracy of the temperature profile, the $CO_2$ profile retrieval with GFIT2 presents no clear advantage over scaling retrievals for the purpose of ascertaining the total column.

## 1. Introduction

Carbon dioxide ($CO_2$) is the most abundant well-mixed greenhouse gas in the atmosphere and the main driver of the increase in global mean surface temperatures since the start of the industrial era (Ciais et al., 2013; Myhre et al., 2013). A yearly global carbon budget has been produced by the Global Carbon Project since 2012 (Friedlingstein et al., 2019; Le Quéré et al., 2013,

2014, 2015b, 2015a, 2016, 2018b, 2018a). It presents current knowledge of $CO_2$ emissions to inform policies that aim to reduce the emissions of greenhouse gases into the atmosphere. The project uses ensembles of models and inventories, as well as $CO_2$

surface measurements, to estimate different components of the global emissions of $CO_2$. It also uses $CO_2$ fluxes obtained from atmospheric inversions (Chevallier et al., 2005; van der Laan-Luijkx et al., 2017; Rödenbeck et al., 2003; Saeki and Patra, 2017) as a semi-independent validation tool for these estimates, most of the $CO_2$ measurements used in these inversions come from surface networks. Since 2014, the project makes mention of the potential of inversions using space-based measurements of total column $CO_2$ to provide additional constraints on sources and sinks of $CO_2$.

Column-averaged dry-air mole fractions of $CO_2$ ($XCO_2$), are retrieved from solar absorption spectra measured from space by the Greenhouse gases Observing SATellite (GOSAT, and GOSAT-2) (Kuze et al., 2009, 2016; Nakajima et al., 2012), the Orbiting Carbon Observatory (OCO-2, and OCO-3) (Crisp, 2008, 2015; Eldering et al., 2019), and Tansat (Liu et al., 2018). $CO_2$ fluxes obtained from inversions assimilating OCO-2 observations over land are now becoming as reliable as those

obtained from inversions using surface air sampling networks (Chevallier et al., 2019). Measurements of $XCO_2$ by satellites can be made with unprecedented spatial coverage. Inversions using $CO_2$ total columns over land are less sensitive to transport errors than inversions using surface $CO_2$ (Basu et al., 2018; Rayner and O'Brien, 2001), which requires accurate modeling of the planetary boundary layer height and vertical mixing, both of which are a major source of uncertainty in inversions (Parazoo et al., 2012). However, even small (< 1 ppm) spatially coherent biases in column measurements can have a large impact on

inversions assimilating $XCO_2$ (Chevallier et al., 2007), and efforts must be made to characterize and minimize such biases (Kiel et al., 2019; O'Dell et al., 2018).

The Total Carbon Column Observing Network (TCCON) is a ground-based network of high-resolution (0.02 cm$^{-1}$) ground-based Fourier transform Infrared (FTIR) spectrometers that record Short Wave IR (SWIR) solar absorption spectra (Wunch et

al., 2011b). TCCON produces retrievals of $XCO_2$ is retrieved and widely used to validate satellite observations and to study the carbon cycle (Wunch et al., 2010a, 2017; Keppel-Aleks et al., 2012, 2013). New versions of the TCCON retrieval algorithm (GGG) are released every few years, and each new version is designed to improve the quality of the data.

GGG2014 (Wunch et al., 2015) is the current version of the GGG software used by TCCON to transform measured

interferograms into spectra, and then to retrieve trace gas mixing ratios from those spectra. Central to this process is GFIT, a non-linear least-squares spectral fitting algorithm. A forward model computes an atmospheric transmittance spectrum using a priori knowledge of atmospheric conditions. An inverse method then compares the measured spectrum with the resulting calculation and adjusts the retrieved parameters to obtain the best fit. In GFIT, these parameters include volume mixing ratio scaling factors (VSF) for the different fitted gases. GFIT performs profile scaling retrievals: for each retrieved trace gas, a

single VSF scales the entire a priori concentration profile at all altitude levels simultaneously and therefore the retrieved profile shape is unchanged from the a priori profile shape. Scaling retrievals do not require inter-level constraints on a priori





concentration uncertainties. In GFIT, the a priori VSF value of the main target gas in a spectral window is 1 with an uncertainty of $10^6$, and $XCO_2$ can be retrieved with a 2-σ precision and accuracy of 0.8 ppm (Wunch et al., 2010). GFIT minimizes the spectral fit residuals: the difference between the measured and calculated spectra. The measurement noise is not required to be accurately known; all retrievals from TCCON $CO_2$ windows use an assumed signal-to-noise ratio (SNR) of ~200. This assumption has only a small effect on the result because for $CO_2$ the absorption line depths and the spectral fitting residuals far exceed the measurement noise.

Even though TCCON $XCO_2$ observations are precise and accurate, they explicitly lack information about the vertical distribution of $CO_2$ in the atmosphere, which is of interest for the validation of satellite measurements and model simulations, and could improve the ability of atmospheric inversions to resolve emissions at regional scales (Keppel-Aleks et al., 2011). The most precise and accurate source of information on $CO_2$ vertical profiles are provided by air samples collected at different altitudes using weather balloons or aircrafts, but these observations are sparse in space and time. Aircraft vertical profiles are used as validation tools for inversion studies (Peters et al., 2007; Stephens et al., 2007; Pickett-Heaps et al., 2011), which requires them to remain independent from the inversion systems (Chevallier et al., 2019). Obtaining reliable $CO_2$ profile information from ground-based direct sun measurements could significantly augment the number of observations available for verification and assimilation in atmospheric inversions. Vertical profile information derived from ground-based absorption spectra cannot be as accurate as aircraft-based vertical profiles, and would also be spatially sparse, but would provide a higher temporal sampling.

$CO_2$ profile retrievals from ground-based SWIR spectra have been calculated using the band centered at 1.6 µm with a Voigt line shape (Kuai et al., 2012), and in the band centered at 2.06 µm with the PROFFIT optimal estimation software package (Hase et al., 2004) fitted with a Voigt line shape with line mixing (Dohe, 2013). In our approach, we use the GFIT2 software package initially described by Connor et al. (2016), which is a profile retrieval algorithm based on the GGG software suite, but modified such that it allows the profile shape to vary during the retrieval process. Instead of retrieving a single VSF value that scales the whole a priori profile, a VSF value is retrieved for each atmospheric level. The algorithm thus has much more freedom to fit the observed spectra but is also more sensitive to uncertainties in the forward model calculations such as errors in the atmospheric temperature profile, spectroscopic errors, and instrument misalignment, for example.

Connor et al. (2016) showed that $CO_2$ profile retrievals in the $CO_2$ band centered at 1.6 µm are very sensitive to errors in spectroscopy. GFIT2 was first developed using the GGG2014 version of the GGG suite (Wunch et al., 2015), which uses a Voigt line shape to compute absorptions coefficients. In this study, we use the GGG2020 version, which will be released in early 2021. This version of the code implements quadratic speed-dependent Voigt line shapes with line mixing (qSDV+LM) for $CO_2$ (Mendonca et al., 2016) and $CH_4$ (Mendonca et al., 2017) bands, and qSDV line shapes for $O_2$ in the band centered at 1.27 µm (Mendonca et al., 2019). This leads to significantly better spectral fits, especially in the strong $CO_2$ band centered at 2.06 µm, and smaller variations of gas amount with airmass. Other improvements to the forward model include: (1) updates





to the spectroscopic linelist (Toon, 2015); (2) a solar gas stretch fitted to account for Doppler-driven differences between solar and telluric wavenumber scales; and (3) improved a priori profiles as described in Sect. 2.2.

This study assesses the quality of $CO_2$ profile retrievals with GFIT2 implemented in GGG2020. Section 2 describes the retrieval algorithm and our methodology. Section 3 presents a sensitivity study using synthetic spectra, followed by retrievals

using real measured spectra. Finally, Sect. 4 presents a summary of the results and conclusions.

## 2.    Methods

In this study, GFIT2 is used to retrieve $CO_2$ profiles from the two original TCCON spectral windows and three new windows that possess a large range of opacities, and therefore vertical sensitivities. These windows are presented in Table 1 and Fig.1. The TCCON1 window (centered at 6220 $cm^{-1}$) and TCCON2 window (centered at 6339.5 $cm^{-1}$) are used to derive $XCO_2$ in

the public TCCON data products, because the spectral absorption lines opacities are close to 1 and are therefore equally sensitive at most altitudes. The $CO_2$ line intensities in the weak windows are 10 times smaller than in the standard TCCON windows, providing more sensitivity to $CO_2$ variations aloft. The $CO_2$ lines in the strong window are 15 times stronger than those in the standard TCCON windows, providing more sensitivity to $CO_2$ variations near the surface. All windows have an average lower-state energy (E'') of roughly 240 $cm^{-1}$, rendering the retrieved total column of $CO_2$ highly independent of the

assumed temperature (<0.1%.$K^{-1}$). The derivation of $XCO_2$ as calculated in GGG is described in Appendix A. XCO2 is the ratio of the $CO_2$ column to the column of dry air, and the column of dry air is expressed as the retrieved $O_2$ column (from the window centered at 7885 $cm^{-1}$, see Table 1) divided by 0.2095 (Wunch et al., 2011b).

**Table 1: $CO_2$ spectral windows used with GFIT2. Interfering absorbers labeled "solar" are due to absorption by heavy metal ions**
**(e.g., Fe, Si, Ca, Ni) in the solar photosphere. Also shown are the strength-weighted averages of the lower-state energy (E''), and of the line strengths (S) over all the $CO_2$ lines in each window. The column of $O_2$, retrieved with scaling retrievals from the $O_2$ window, is used to compute $XCO_2$.**

| Window name | center ($\mu$m) | center ($cm^{-1}$) | width ($cm^{-1}$) | Primary interfering absorbers | E'' ($cm^{-1}$) | S ($cm^{-1}$/(molecule.$cm^{-2}$)) $\times 10^{-23}$ |
|---|---|---|---|---|---|---|
| TCCON1 | 1.61 | 6220 | 80 | solar, $H_2O$ | 245.3 | 1.14 |
| TCCON2 | 1.58 | 6339.5 | 85 | solar, $H_2O$ | 254.6 | 1.14 |
| Weak1 | 1.65 | 6074 | 70.8 | $CH_4$, solar, $H_2O$ | 223.5 | 0.118 |
| Weak2 | 1.54 | 6499.1 | 69.8 | solar, $H_2O$, HDO | 229.3 | 0.130 |
| Strong | 2.06 | 4852.87 | 86.26 | $H_2O$, $^{13}CO_2$, solar | 243.8 | 17.8 |
| $O_2$ | 1.27 | 7885 | 240 | solar, $H_2O$, HF, $CO_2$ | 203.4 | 0.00518 |





A qualitative representation of the vertical sensitivity due to the range of different line opacities is presented in Fig. 2 which shows the normalized $CO_2$ Jacobian for typical absorption lines in the Strong window (centered at 4852.87 $cm^{-1}$), the Weak1 window (centered at 6074 $cm^{-1}$), and the TCCON1 window (centered at 6220 $cm^{-1}$). The strong saturated lines of the Strong window are more sensitive to levels below 5 km than in the TCCON1 window, but the Strong window also contains lines of intermediate absorption strength that provide more uniform sensitivity up to ~10 km, and that extend the window's sensitivity

to up to 30–40 km. The saturated lines in the Strong window correspond to the 20013–00001 band, while the lines of intermediate strength around 4820 $cm^{-1}$ come from the R-branch of the 21113–01101 band. The TCCON1 window has more uniform sensitivity up to ~10–15 km and contains weak lines which contain information on $CO_2$ above 15 km. The Weak1 window is less sensitive below 10 km and has more uniform sensitivity between 10–20 km. Figure 2 also shows little to no sensitivity to levels above ~30 km in all windows.


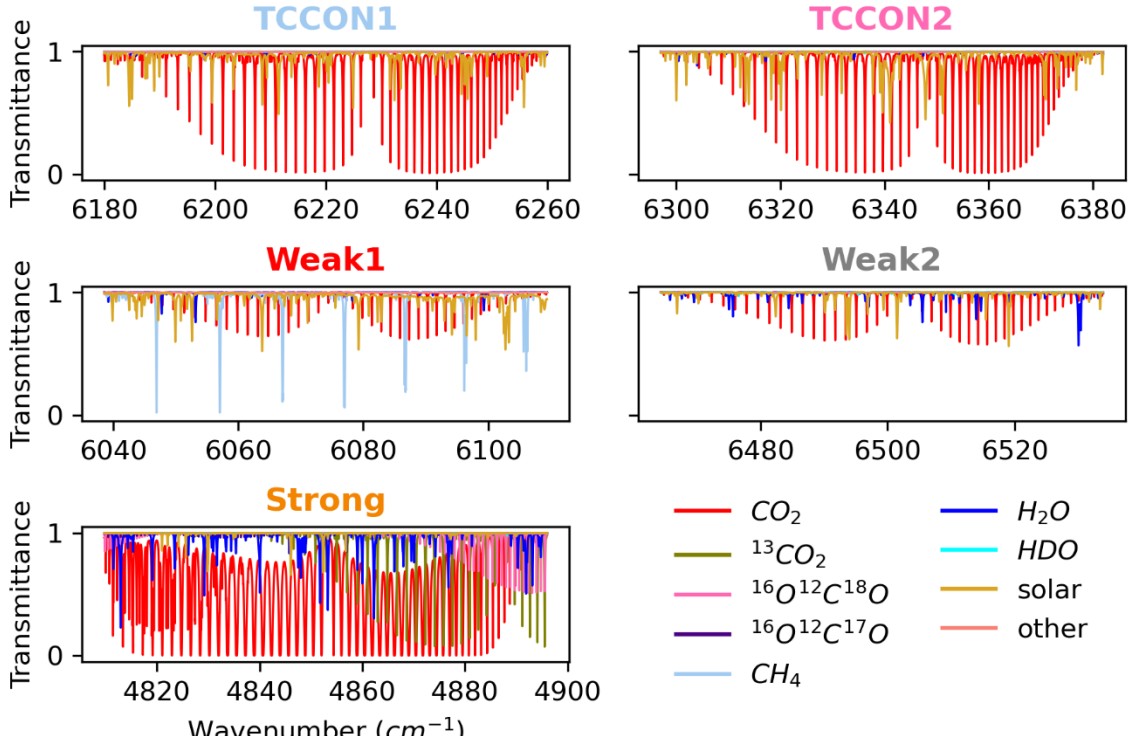

**Figure 1: Contributions of different absorbing gases to the calculated transmittance spectrum on a dry winter day at a solar zenith angle of 60.6° for each of the spectral windows used to retrieve $CO_2$.**



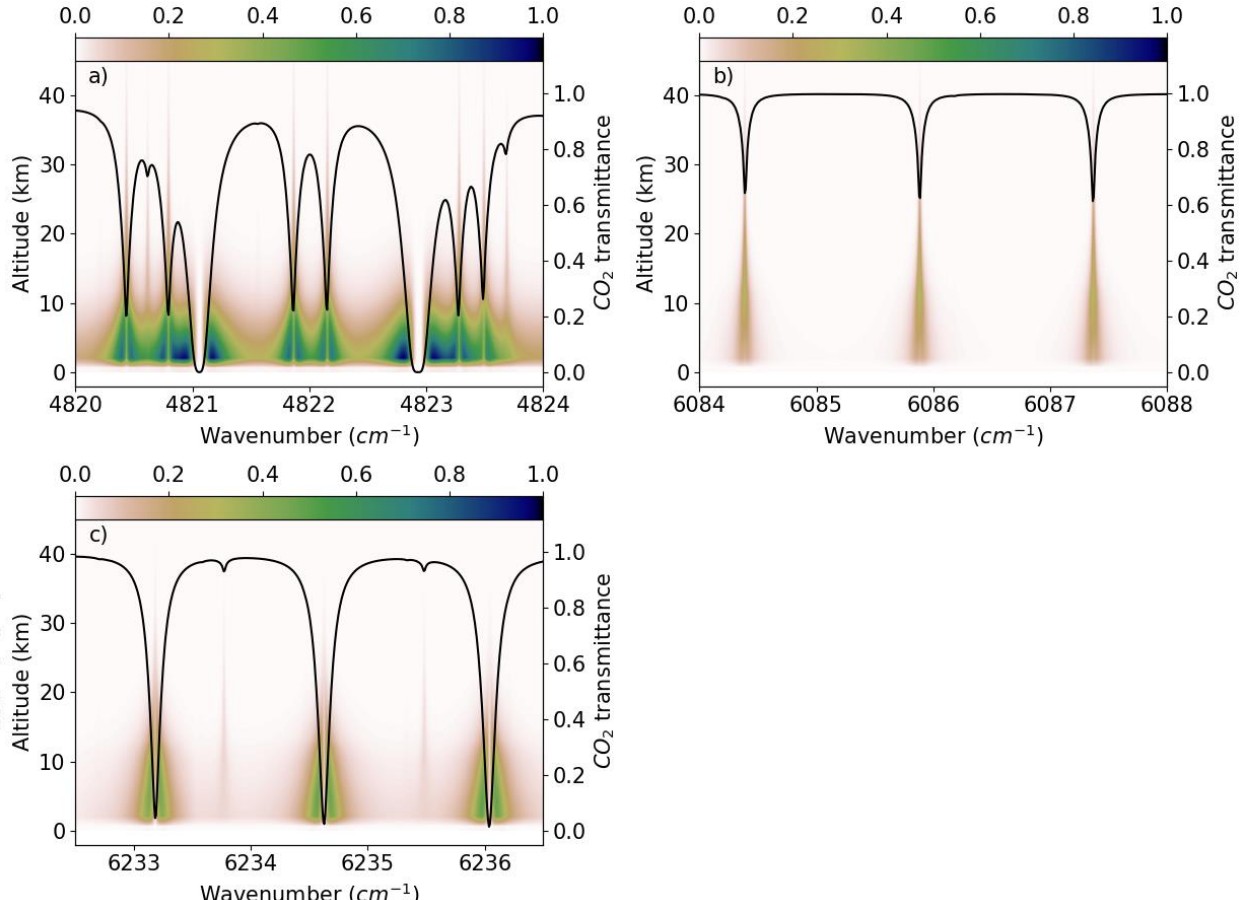

**Figure 2: CO₂ absorption lines (black line) overlaid on heatmaps of the CO₂ Jacobian for lines of (a) the Strong window; (b) the Weak1 window; and (c) the TCCON1 window. The color bar represents the normalized Jacobian where 1 corresponds to the maximum amongst all the CO₂ Jacobians from the five CO₂ windows. Lines of the Weak2 and TCCON2 windows are not shown as they look like the Weak1 and TCCON1 windows, respectively.**

### 2.1 Retrieval algorithm

The GFIT2 retrieval algorithm is described in detail in Appendix B and follows the formulation of Rodgers (2000). Currently, GGG has no option to simultaneously retrieve information about a gas from spectral windows that are not contiguous in wavenumber. Therefore, we retrieve trace gas information from each window separately. We see no advantage to fitting non-contiguous windows in parallel, rather than in series, and then averaging the results. Table 2 summarizes the components of the state vector used in GFIT2. Fifty-one VSFs are retrieved (one for each atmospheric level) for the primary target gas, while only one VSF is retrieved for each of the interfering species profiles. Aside from the retrieved gases, other fitted parameters are part of the state vector. Orthogonal continuum basis functions are used to fit the shape of a spectrum's continuum, with different orders of curvature. An overall frequency stretch is retrieved for all lines and a second stretch is retrieved to correct





for differences between the solar and telluric wavenumber scales. A zero-level offset is also retrieved in the Strong window that makes use of saturated lines.


**Table 2: Components of the state vector in GFIT2 profile retrievals.**

| State vector parameter | Number of elements |
|---|---|
| Main target gas ($CO_2$) | 51 (number of atmospheric levels) |
| Interfering species | 3–6 (scaling retrievals) |
| Continuum basis functions: | N (5 in the Strong window, 3 in the other windows) |
| Continuum level | 1 |
| Continuum tilt | 1 |
| Continuum curvature | N-2 |
| Frequency shift | 1 |
| Solar-gas stretch | 1 |
| Zero-level offset | 0 (1 in the Strong window) |

In principle, a $CO_2$ profile retrieval should have little sensitivity to errors in the a priori $CO_2$ profile (differences from the true profile) since it can adjust for differences between measured and calculated spectra caused by erroneous prior profile shapes

(Connor et al., 2016). However, the retrieval may also conflate errors due to other sources, such as incorrect spectroscopic parameters, incorrect modeling of the instrument line shape, or errors in the a priori meteorology and profiles of interfering species, with these errors in the a priori $CO_2$ profile.

**2.2 Data sets**

$CO_2$, and $CH_4$ a priori profiles were built by combining the balloon-borne AirCore (Karion et al., 2010) profiles with surface

in situ measurements, adding the GGG2020 a priori profile above the maximum altitude sampled by AirCore. These composite profiles will be referred as "truth". The $CH_4$ profile is included because $CH_4$ is an interfering gas in the Weak1 window. AirCore is a sampling system that consists of a long, coiled stainless-steel tube initially filled with a dry calibrated gas. As a balloon carries it up, the fill gas evacuates. When the AirCore descends from the stratosphere, ambient air enters the tube through the open end. Upon landing, the AirCore is quickly retrieved for subsequent laboratory analysis, wherein the sample

is pushed through a continuous gas analyzer. The first gases to come out were the last to enter, and vice versa, allowing the preserved atmospheric trace gas concentration profiles to be derived. This method has precision similar to, or better than, discrete gas flask samples, with a repeatability of 0.07 ppm for $CO_2$ concentrations (Karion et al., 2010). The balloons reach ~30 km altitude, with profiles retrieved to ~25km and therefore sample 98% of the mass of the atmosphere. In Sect. 3.2, AirCore profiles from the v20181101 dataset were used as "truth" to assess the quality of GFIT2 profile retrievals. We used

all AirCore profiles measured over the Lamont TCCON station that had coincident ground-based measurements within ± 1 h





of the AirCore landing and within ±1.5 h of the closest a priori time. All figures showing profiles use the average of profiles retrieved from the coincident spectra. The launch dates of the eight AirCore profiles used are presented in Table 3. An iMet-1 radiosonde carried by the same balloon as the AirCore provides in situ temperature and relative humidity profiles.

**Table 3: AirCore launch dates and number of coincident spectra within ±1 h of the AirCore last sampling time and within ±1.5 h of the closest a priori time. The range of solar zenith angles covered by the coincident spectra is also shown.**

| Launch date | Coincident spectra | Solar Zenith angles (degrees) |
|---|---|---|
| 14 January 2012 | 66 | 60.6–73.8 |
| 15 January 2012 | 48 | 65.6–77.9 |
| 23 July 2013 | 45 | 20.8–36.5 |
| 26 February 2014 | 62 | 46.6–59.0 |
| 27 February 2014 | 42 | 46.2–53.3 |
| 17 September 2014 | 49 | 37.9–51.1 |
| 19 October 2016 | 31 | 47.1–50.3 |
| 11 April 2017 | 34 | 31.2–39.2 |

Instead of the diagonal prior covariance used in Sect. 3.1, a more realistic $CO_2$ prior covariance matrix was built for retrievals with real spectra in Sect. 3.2. The difference between GGG2020 a priori $CO_2$ profiles and aircraft profiles (Biraud et al., 2013)

over Lamont from NOAA's ObsPack (Sweeney et al., 2017) between 500 and 5000 m were computed for 382 aircraft profiles and for each month between 2008 and 2016. The mean difference profile plus one standard deviation of the month with the largest differences, August, was used to build the diagonal of the a priori covariance matrix. The a priori $CO_2$ uncertainty can be expressed as:

$$\sigma_i = 3.99 e^{-0.92 x_i} + 0.98 \tag{1}$$

where $x$ is the altitude of the $i^{\text{th}}$ atmospheric level in kilometers. The a priori covariance is expressed as:

$$\mathbf{z}_{i,j} = \mathbf{x}_i \tag{2}$$

$$(\mathbf{\Delta z})_{i,j} = \left| \mathbf{z}_{i,j} - \mathbf{z}^T_{i,j} \right| \tag{3}$$

$$(\mathbf{S}_a)_{i,j} = (\mathbf{\sigma}^T \mathbf{\sigma})_{i,j} \times e^{-\frac{(\Delta z)_{i,j}}{h}} \tag{4}$$

where $z$ is a matrix with each row containing the altitude profile, $\mathbf{\Delta z}$ is the matrix of absolute altitude differences between each

level, $\mathbf{S}_a$ is the a priori covariance matrix, and $h$ is the length scale of interlayer correlations. The length scale was set to 2 km based on the width of the rows of correlation matrices built from the ensemble of aircraft vertical profiles.

The vertical grid used in the retrievals presented in this study has 51 levels between 0 and 70 km, and the spacing between levels increases with altitude (see Appendix A). Since the AirCore profiles do not extend down to the surface or above about



25 km, other sources are used to complete the "true" $CO_2$ profile. The TCCON spectrometer used in this study is located at the U.S. Department of Energy Atmospheric Radiation Measurement program (ARM) central facility in Lamont, Oklahoma. The facility hosts a suite of instruments for remote and in situ measurements of the atmosphere. When available within 5 h of the last AirCore sampling time, surface $CO_2$ and $CH_4$ measurements from precision gas systems were used (Biraud and Moyes,

2001). When they were not available, measurements from discrete flask samples were used (on 23 July 2013, 27 February 2014, and 17 September 2014) (Biraud et al., 2002). Surface pressure, temperature and relative humidity were obtained from in situ measurements at the Lamont central facility.

GGG2020 uses 3-hourly a priori profiles of the atmospheric state. For each spectrum in the retrievals, GGG uses the nearest a priori profile in time. The a priori meteorology and $H_2O$ profiles are obtained from analyses of the Global Modeling and

Assimilation Office (GMAO) Goddard Earth Observing System Version 5 Forward Processing for Instrument Teams (GEOS5-FPIT) (Lucchesi, 2015). The $CO_2$ a priori profiles are constructed from the deseasonalized NOAA Mauna Loa and Samoa flask data (Dlugokencky et al., 2019) by determining the transport lag between the measurement site and each level of the a priori (Laughner et al., n.d.). In the troposphere, this is done with an age-of-air formula and an effective latitude that accounts for synoptic motion of air. In the stratosphere, this is obtained from an age climatology derived from a Chemical Lagrangian

Model (McKenna, 2002) of the stratosphere using equivalent latitude to account for air motion. The stratospheric priors also account for turbulent mixing with age spectra (Andrews et al., 2001). A seasonal cycle parametrization is then applied and the resulting $CO_2$ profiles are corrected to match the $CO_2$ latitudinal gradients observed by the High-Performance Instrumented Airborne Platform for Environmental Research (HIAPER) Pole-to-Pole Observations (HIPPO) (Wofsy, 2011), and by the Atmospheric Tomography (ATom) mission (Wofsy et al., 2018).

**2.3 Information content and degrees of freedom**

The information content in the profile retrieval can be quantified using the averaging kernel matrix $\mathbf{A}$ (Rodgers, 2000). The information content $H$ is defined as:

$$H = -\frac{1}{2} ln(|\mathbf{I} - \mathbf{A}|),$$ (5)

where "ln" is the natural logarithm and $|\mathbf{I} - \mathbf{A}|$ is the determinant of the difference between the identity matrix and the

averaging kernel matrix. The degrees of freedom for signal (DOFS) can be expressed as:

$$DOFS = tr(\mathbf{A}).$$ (6)

The DOFS can be divided into the $CO_2$ profile DOFS and the DOFS corresponding to the rest of the state vector elements. The profile DOFS can be interpreted as the number of independent pieces of information that improve the retrieved $CO_2$ profiles compared to the a priori.



### 3.  Results

In Sect. 3.1, we investigate the sensitivity of the profile retrievals to different sources of error using synthetic spectra produced by running the GGG forward model with a given set of atmospheric conditions. The resulting spectra were then used as input to the profile retrieval algorithm using the same set of atmospheric conditions, except for a perturbation in either the $CO_2$, temperature, or $H_2O$ profiles, or in the spectroscopic parameters of $CO_2$ lines (air- and self-broadened half-width coefficients, and their temperature dependence). In these retrievals, the SNR of the spectrum to be fitted is set to 1000 and the $CO_2$ a priori covariance matrix is diagonal with 5% (~20 ppm) uncertainty at all levels. No noise is added to the calculated spectra, but the assumed 1000:1 SNR is used to build the measurement covariance matrix and affects the relative weight of the measurement and the a priori. The weak prior constraint and high SNR serve to highlight the sources of variability in the retrieved profiles.

In Sect. 3.2, $CO_2$ profile retrievals are tested with atmospheric solar absorption spectra measured at the Lamont, Oklahoma (USA) TCCON site. If the forward model were perfect and the a priori state equal to the true state of the atmosphere, the retrieved scale factor at each level would be equal to 1. However, errors in the forward model (including spectroscopy, a priori meteorological information, radiative transfer, and instrument line shape) cause the retrieved scale factors to deviate from 1. To isolate the effect of instrument misalignment and errors in spectroscopic parameters from errors in a priori meteorology, we build a priori profiles of $H_2O$, temperature, $CO_2$ and $CH_4$ using in situ measurements. In Sect. 3.2 we also use an a priori covariance matrix with off-diagonal elements based on comparisons between the a priori profile and aircraft profiles, as described in Sect. 2.2.

**3.1. Synthetic spectra**

In this section, we attempt to identify the main sources of error in the retrieved $CO_2$ profiles. To do this we use synthetic spectra that are calculated with GFIT's forward model for a given set of inputs (atmospheric conditions and spectroscopic parameters). These "perfect" synthetic spectra are then used as measurements to be fitted in retrievals with one perturbed input. Thus, when the perturbed input is not the a priori $CO_2$ profile itself, the a priori $CO_2$ profile is the "truth". In Sect. 3.1.1, we look at the ability of the retrieval algorithm to retrieve $CO_2$ when it is the only unknown.

Over the course of a day, the water vapour profile can vary by 40% and the temperature profile can vary by more than 10°C in the lowest troposphere, and therefore 3-hourly a priori meteorological information could differ from the true atmospheric state by several degrees C for temperature and by 10% for water vapour. In Sect. 3.1.2, we perturb the a priori $H_2O$ profile, the main interfering absorber. In Sect. 3.1.3, we perturb the temperature profile, as the intensity and width of all absorption lines depend on temperature. Finally, in Sect. 3.1.3 we perturb spectroscopic line parameters themselves to within their uncertainties.



### 3.1.1 Perturbed CO₂ profile

With a perturbed $CO_2$ prior profile, the algorithm can retrieve the true profile shape very well in all windows, even with an a priori profile vastly different from the truth as shown in Fig. 3. In Fig. 3(a), when using the same prior that generated the synthetic spectrum, the retrieved profiles do not align exactly with the prior profile. This is due to small imperfections in the synthetic spectra, but these result in differences of less than 1 ppm at any altitude. In Fig. 3(c) the standard GGG2020 a priori is used as the a priori, while the "true" $CO_2$ profile used to generate the synthetic spectrum was built from a composite "true" profile as described in Sect. 2.2. In each window the retrieved profile is within 2 ppm of the truth. In Fig. 3(e) a constant $CO_2$ profile with 380 ppm at all levels is used as the a priori. Again, the retrieved profiles are within 2 ppm of the truth except at the bottom and top of the profile where most of the information comes from the a priori. This self-consistency test shows that the GFIT2 algorithm works as expected and can accurately retrieve $CO_2$ when the a priori $CO_2$ profile is the only source of uncertainty.





**Figure 3: The left-hand panels show CO₂ profiles retrieved using synthetic spectra. In (a), we use the AirCore profile, which was used to generate the synthetic spectra, as the a priori. In (c), we use the GGG2020 a priori CO₂ profile as the a priori profile. In (e), we use a constant CO₂ a priori profile. The right-hand panels: (b), (d), and (f), show the difference between the retrieved profiles and AirCore, corresponding to (a), (c), and (e) respectively.**





### 3.1.2 Perturbed H₂O profile

Figure 4 shows the effect of a +10% perturbation to the H$_2$O vapour profile below 5 km for a dry winter day and a wet summer

day. It leads to 2 ppm deviations from the CO$_2$ a priori in the Strong window under dry conditions and up to 15 ppm under wet

conditions. In both cases, the deviations from the truth in the CO$_2$ profiles retrieved from the other windows were within 2

ppm.

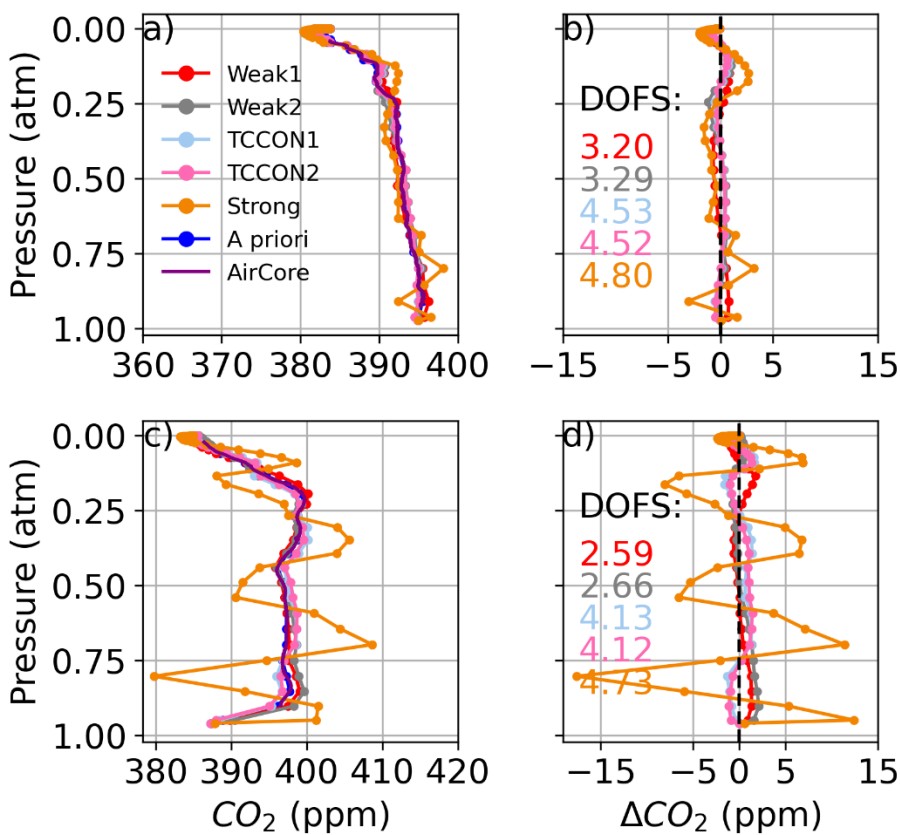

**Figure 4: The left-hand panels show CO₂ profiles retrieved using synthetic spectra. 10% is added to the H₂O profile below 5 km for**
**(a) dry conditions on 14 January 2012, and for (c) wet conditions on 23 July 2013. The right-hand panels: (b) and (d), show the**
**difference between the retrieved profiles and AirCore, corresponding to (a) and (c) respectively.**

### 3.1.3 Perturbed temperature profile

A +5°C perturbation to the temperature profile below 5 km (0.5 atm < P < 1.0 atm), as in Fig. 5(a), leads to deviations from

the truth in the retrieved CO$_2$ profiles of up to 50 ppm in the Weak and TCCON windows, and up to 100 ppm in the Strong

window. In that case the fit residuals can exceed 1% in the Strong window and 0.5% in the TCCON windows. For the retrievals

used to obtain the profiles in Fig. 5(a) the SNR was set to 100 in the Strong window, 200 in the TCCON windows, and 1000

in the Weak windows. In Fig. 5(c) and (e) the SNR is set to 1000 in all windows. In Fig. 5(c), a +2°C perturbation is applied



between 8 and 13 km ($0.2\,\text{atm} < P < 0.35\,\text{atm}$). The amplitude of deviations in the TCCON windows and in the Strong window is close to 50 ppm at ~0.9 atm and 100 ppm at ~0.2 atm. In the two Weak windows, the deviation amplitude is ~10 ppm at

~0.9 atm and ~20 ppm at 0.2 atm. In Fig. 5(e), a +2°C perturbation is applied above 15 km. In the Strong window, the resulting deviation at pressures > 0.6 atm has the smallest amplitude amongst the five windows, within 4 ppm, and the deviation at ~0.2 atm is ~20 ppm. In the TCCON windows, the deviation at pressures > 0.6 atm is reduced to ~10 ppm while the deviation at pressures > 0.6 atm is comparable to that in Fig. 5(b). In the two Weak windows, the deviations at ~0.9 atm is unchanged when to compared to Fig. 5(b) and the deviation at ~0.2 atm is reduced from ~15 ppm to ~10 ppm.



**Figure 5: The left-hand panels show CO₂ profiles retrieved using synthetic spectra for: (a) +5°C added to the a priori temperature profile below 5 km, (c) +2°C between 8 and 13 km, and (e) +2°C above 15 km. The right-hand panels: (b), (d), and (f), show the difference between the retrieved profiles and AirCore profile, corresponding to (a), (c), and (e) respectively. Note the difference in the horizontal axis range between the panels. Here 5 km corresponds to ~0.55 atm, 8–13 km to ~0.36–0.17 atm, and 15 km to ~0.125 atm.**





From the results in Sect. 3.1.1, 3.1.2, and 3.1.3, we observe that $CO_2$ profile retrievals do not need accurate prior knowledge of the $CO_2$ profile, but require accurate knowledge of the prior temperature and water vapour profiles. Moreover, these results
suggest that errors in the temperature profile are the main source of deviations from the truth in retrieved $CO_2$ profiles. Retrievals using the two Weak windows are the least affected by biases in the prior temperature and water vapour profiles. The need for accurate a priori water vapour profile could be alleviated by retrieving $H_2O$ profiles simultaneously with $CO_2$ profiles, but this was not tested with GFIT2 which currently can only retrieve the main target gas in a window with profile retrievals. In addition, $H_2O$ profile retrievals would also be affected by temperature errors.

**3.1.4 Perturbed line parameters**

The linelist used by GGG is a compilation of different versions of the HITRAN linelists (Gordon et al., 2017; Rothman et al., 2005, 2009, 2013; Toon, 2015; Toon et al., 2016). GGG2020 has the option to use either the qSDV+LM line shape or the Voigt line shape for some windows and gases (Mendonca et al., 2016, 2017, 2019). The reference linelists and the uncertainties on air- and self-broadened Lorentz half-width coefficients, and their temperature dependence, are summarized in Table 4. The
qSDV+LM line shape is only implemented for the $CO_2$ lines of the two TCCON windows and the Strong window, for the $CH_4$ lines of the Weak1 window, and for the $O_2$ lines of the oxygen window centered at 7885 cm$^{-1}$. The qSDV+LM line shape is not implemented for the $CO_2$ lines of the Weak1 and Weak2 windows, but these weak lines are minimally affected by line mixing, and they lack laboratory measurements of speed-dependent line parameters. The effect of errors in the half-width coefficients on the retrieved $CO_2$ profiles was tested by increasing both the self- and air-broadened Lorentz half-width
coefficients by 0.1% for all $CO_2$ lines as shown in Fig. 6(a). This perturbation corresponds to the median uncertainty of these parameters in the Strong and TCCON windows as shown in Table 4. This caused deviations of up to 10 ppm in the Strong window, 5 ppm in the TCCON windows, and 2 ppm in the Weak windows. Similar deviations are obtained by perturbing the temperature dependence of the half-width coefficients by -1% as shown in Fig. 6(b). In this case, the deviations appear mirrored about the a priori compared to Fig. 6(a). The shape of deviations in both cases is similar; it is also similar to the shape obtained
in Fig. 5 from perturbing the temperature profiles. This is because all those perturbations ultimately lead to an altered line width and all cause residuals patterns that cannot be distinguished from each other, as illustrated in Fig. 7. This implies that errors in the a priori temperature profile, water vapour profile, and spectroscopic widths are difficult to disentangle in the current GFIT2 profile retrieval. A simultaneous temperature (hence pressure) and $CO_2$ profile retrieval would be necessary to overcome these issues.


A factor 10 increase in the perturbations applied to the width coefficients or their temperature dependence also leads to a factor 10 increase in the amplitude of deviations in the retrieved $CO_2$ profiles. Panels (a) and (b) of Fig. 6 use perturbations corresponding to uncertainties in the line parameters when using qSDV+LM for the TCCON windows and the Strong window.



The same perturbations were applied for all five windows. However, in the Weak1 and Weak2 windows, these perturbations

are 10 times smaller than realistic uncertainties as reported in Table 4 for the Voigt line shape. Therefore, for the Weak windows, we can expect deviations from the truth 10 times larger than in Fig. 6, within ~10–20 ppm.

**Table 4: 1-σ relative errors of the air- and self-broadened Lorentz half-width coefficients (b) and of their temperature dependence (n). The values from Benner et al. (2016) and Devi et al. (2007a,b) use the median 1-σ uncertainty for the whole band, from the**

**Appendix or supplemental files of these studies. The values for the Voigt line shape use the error codes reported in the HITRAN2016 linelist (Gordon et al., 2017).**

| Line shape | Window (band) | b (air) (%) | n (air) (%) | b (self) (%) | n (self) (%) | Reference |
|---|---|---|---|---|---|---|
| Voigt | TCCON1 | >=1 and <2 | - | >=1 and <2 | - | Toth et al. (2008) |
| | TCCON2 | | | | | |
| | Weak1 | | From <10 To <1 | | | Lamouroux et al. (2015) |
| | Weak2 | | | | | Gordon et al. (2017) |
| | Strong | | | | | |
| qSDV+LM | TCCON1 (30013–00001) | 0.13 | - | 0.07 | | Devi et al. (2007a) |
| | TCCON2 (30012–00001) | 0.14 | | 0.07 | | Devi et al. (2007b) |
| | Strong (20013–00001) | 0.03 | 0.12 | 0.09 | 0.33 | Benner et al. (2016) |
| | Strong (21113–01101) | 0.25 | 1.47 | 0.49 | 2.27 | |

In Connor et al. (2016), the authors used a Voigt line shape. Figure 6(e) shows the effect of fitting with a Voigt line shape a synthetic spectrum that was generated using qSDV+LM. In that case the fit residuals in the Strong window can exceed 1% and

the residuals in the TCCON windows can exceed 0.5%. For these retrievals, the SNR is set to 100 in the Strong window, 200 in the TCCON windows, and 1000 in the Weak windows. The profiles retrieved from the Strong window present deviations from the truth within 60 ppm. In the two TCCON windows, the deviations from the truth are within 30 ppm. In the Weak1 window, the deviations from the truth are within 10 ppm, because qSDV+LM was not used to calculate the $CO_2$ line absorptions themselves, but only for the relatively strong $CH_4$ lines in that window. In the Weak2 window, there is no

difference between the two linelists or line shape, and thus the retrieved profile does not differ from the a priori profile. Therefore, even if we assume perfect a priori meteorology, the deviations in the $CO_2$ profiles retrieved from the TCCON1 window observed by Connor et al. (2016), when fitting real spectra could be entirely due to the use of the Voigt line shape.







**Figure 6: The left-hand panels show CO₂ profiles retrieved using synthetic spectra. In (a) the air- and self-broadened half-width coefficients of all CO₂ lines is increased by 0.1%. In (c) the temperature dependence of these coefficients is decreased by 1%. In (e), the synthetic spectrum used as "measurement" is generated with the speed-dependent Voigt line shape with line mixing, but profiles are retrieved using a Voigt line shape. The right-hand panels: (b), (d), and (f), show the difference between the retrieved profiles and AirCore, corresponding to (a), (c), and (e) respectively.**






The effect of the errors in the a priori water vapour and temperature profiles, and in the spectroscopic parameters cannot be mitigated by adjusting the measurement covariance, for example by using a variable SNR. Figure 7 shows an example of spectral residuals from fits to synthetic spectra from the Strong window using scaling retrievals, but with different perturbations applied. Showing residuals from scaling retrievals reveals systematic features that the profile retrieval will attempt to suppress.

Fig. 7(b) presents residuals from fitting a synthetic spectrum using the same a priori that was used to generate the synthetic spectrum. It shows small (< 0.05 %) residuals, caused by the use of a constant ILS across the window for a faster convolution of the spectrum with the ILS. The corresponding profiles are shown in Fig. 3(a). In Fig. 7(c), a 2°C offset is applied to the a priori temperature profile between 8 and 13 km before fitting the synthetic spectrum. In Fig. 7(d), a constant a priori $CO_2$ profile is used to fit a synthetic spectrum that was generated with an AirCore $CO_2$ profile as a priori. In Fig. 7(e), the air- and

self-broadened Lorentz half-width coefficients are increased by 0.1% compared to the parameters used to generate the synthetic spectrum. In Fig. 7(f), the temperature dependence of the air- and self-broadened Lorentz half-width coefficients is decreased by 1% compared to the parameters used to generate the synthetic spectrum. In Fig. 7(g), the GGG2020 a priori meteorology and trace gas profiles are used as a priori profiles instead of the a priori constructed with AirCore profiles used to generate the synthetic spectrum.


In all panels of Fig. 7 except (c) and (g), all the residual features correspond to $CO_2$ absorption lines. In Fig. 7(c), with a perturbation to the a priori temperature profile, there is an added contribution of temperature errors on interfering species. Furthermore, the residuals in Fig. 7(g) result from a combination of errors in the a priori meteorology and trace gas profiles but are dominated by temperature errors. Perturbations in the temperature profile, $CO_2$ profile, or $CO_2$ line width coefficients

all cause residuals with the same shape because they all affect the width of $CO_2$ lines. It is not possible to de-weight the effect of any of those errors by adjusting the measurement error without also losing the ability to correct for residuals caused by $CO_2$ errors. Residuals caused by realistic temperature errors as shown in Fig. 7(c) are of the same magnitude of those caused by unrealistically high errors in the a priori $CO_2$ profile shape as shown in Fig. 7(d).





**Figure 7: Panel (a) shows an example of calculated lines in the Strong CO₂ window. The other panels show residuals from fits to a synthetic spectrum, using the same inputs used to generate the synthetic spectrum except for: (b) no perturbation; (c) +2°C perturbation to the a priori temperature between 8 and 13 km; (d) CO₂ prior profile set to 380 ppm at all levels, corresponding to ~15 ppm offset from the unperturbed prior; (e) air- and self-broadened Lorentz half-width coefficients is increased by 0.1%; (f) temperature dependence of the half-width coefficients decreased by 1%; and (g) using the a priori that would be used by TCCON**
**operational processing, instead of that constructed from in situ measurements, resulting in a combination of different errors in the a priori such as H₂O, temperature, and CO₂. Note the vertical scale of panels (b), (e), and (f) is five times smaller than that of panels (c), (d), and (g).**



### 3.1.5 Synthetic spectra: discussion

For retrievals on synthetic spectra, the "measurement" SNR is set to 1000, which is high compared to most solar spectra measured by TCCON. So in profiles retrieved from real spectra, we can expect a greater influence of the a priori $CO_2$ profile: the deviations will be smaller, and the degrees of freedom for signal will be lower than those shown in the figures of Sect. 3.1. This is not a desirable outcome; the a priori $CO_2$ covariance is meant to nudge the retrieval such that the solution lies close to realistic ensembles of $CO_2$ profiles, not to constrain deviations caused by temperature errors. Tuning the a priori or

measurement covariances is not the right approach until profile deviations caused by typical errors in spectroscopy or meteorology are smaller than typical vertical variations in $CO_2$ profiles. Figure 3 shows that the profile retrieval algorithm works well and could be a powerful tool to derive information about the vertical distribution of $CO_2$, even with ill-defined a priori $CO_2$ profiles. Panels (a) and (b) of Fig. 6 show that profile information could still be retrieved to within ~5 ppm given realistic errors in line width parameters. But as shown with Fig. 5, a temperature retrieval, or correction, is critical to producing

reliable $CO_2$ profile retrievals. This study does not show the effect of typical instrument misalignment errors on the retrieved profiles. GFIT2 currently has no capacity to fit the instrument line shape (ILS) of a misaligned instrument given specific angular and shear misalignments, and instead always assumes a perfect ILS. This is an area of future development for the program.

In Sect. 3.2, GFIT2 is tested with real spectra using an a priori built from in situ measurements. In that case, the deviations from the truth in the retrieved $CO_2$ profile caused by errors in the a priori meteorology (temperature, pressure, and water vapour profiles) are minimized, and the remaining deviations are caused by errors in the spectroscopic line parameters, in the radiative transfer, in the instrument line shape, or in the pointing of the sun tracker.

### 3.2 Real spectra

Here the algorithm is tested with real spectra measured at Lamont as described in Sect. 2.2. A scaling retrieval is performed before each profile retrieval and the root mean square of the residuals from the scaling retrieval is used as measurement uncertainty for the profile retrieval. Since the residuals from the scaling retrieval include systematic features larger than the random noise in the measurement, the root mean square is a conservative estimate of the noise. In Sect. 3.2.1, we present $CO_2$ profiles retrieved from real spectra and we attempt to isolate the effect of errors in instrument line shape, in spectroscopic

parameters, and in pointing, from the effect of errors in meteorology. In Sect. 3.2.2, we present an analysis of the information content and altitude sensitivity of the retrieval. Finally, in Sect. 3.2.3, we compare $XCO_2$ derived from the scaling retrieval to $XCO_2$ derived from the profile retrieval.




### 3.2.1 Profiles

Figures 8 and 9 show $CO_2$ profiles retrieved from real spectra measured from Lamont, OK, on 14 January 2012 and 11 April 2017, respectively. In each figure, panel (a) shows profiles retrieved using in situ profiles (the "truth") as the a priori. In those cases, we assume that deviations from the truth caused by errors in a priori meteorology (pressure, temperature and water vapour profiles) are minimized, and the remaining deviations can be attributed to the combination of instrument misalignment (ILS), pointing errors, or errors in spectroscopic parameters. Panel (c) shows profiles retrieved using the GGG2020 a priori. A first complication for obtaining a satisfactory $CO_2$ profile retrieval is that the a priori $CO_2$ profiles in GGG2020 already compare well with in situ profiles, typically within 5 ppm over Lamont. In Fig. 8(c) and 9(c), the profile that most closely matches the AirCore is the a priori.

Even with ideal prior knowledge of the meteorology and trace gas profiles, the $CO_2$ deviations from the truth can be as large as 50 ppm as shown in Fig. 8(a) and 9(a). When synthetic spectra were perturbed with realistic errors in line width parameters, profile deviations remained within 5 ppm for profiles retrieved from the Strong window and within 10 ppm for the TCCON windows. This suggests that the main cause of deviations in Fig. 8(a) and 9(a) is not due to errors in spectroscopic parameters. The assumption that there is no contribution from temperature errors in the radiosonde profile is supported by the $CO_2$ profile deviation being smallest in the Strong window, which is the most sensitive to temperature errors. The cause of the deviations in Fig. 8(a) and 9(a) could be due to errors in the zero-level offset in the Weak and TCCON windows, where it is assumed to be zero. In these windows the zero-level offset is not fitted as they lack saturated absorption lines. Errors in the instrument line shape, which would affect the line cores, could also contribute to the $CO_2$ profile deviations at higher altitudes.

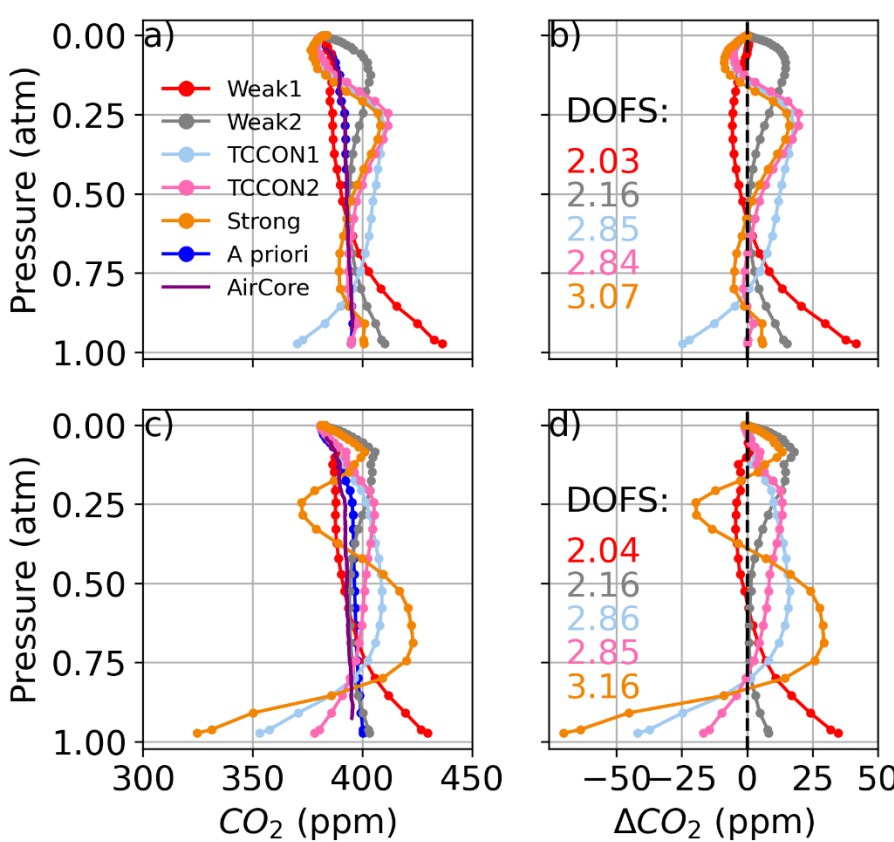

**Figure 8: CO$_2$ profiles retrieved from spectra measured at the Lamont TCCON site on 14 January 2012, at 61–74° solar zenith angle, coincident with AirCore measurements using: (a) the AirCore "truth" as a priori and (c) the GGG2020 a priori. In (b) and (d) the difference of the retrieved profiles minus the AirCore profile is shown, corresponding to (a) and (c), respectively. The points represent the 51 levels of the vertical grid. The DOFS for each retrieval window are indicated in (b) and (d).**



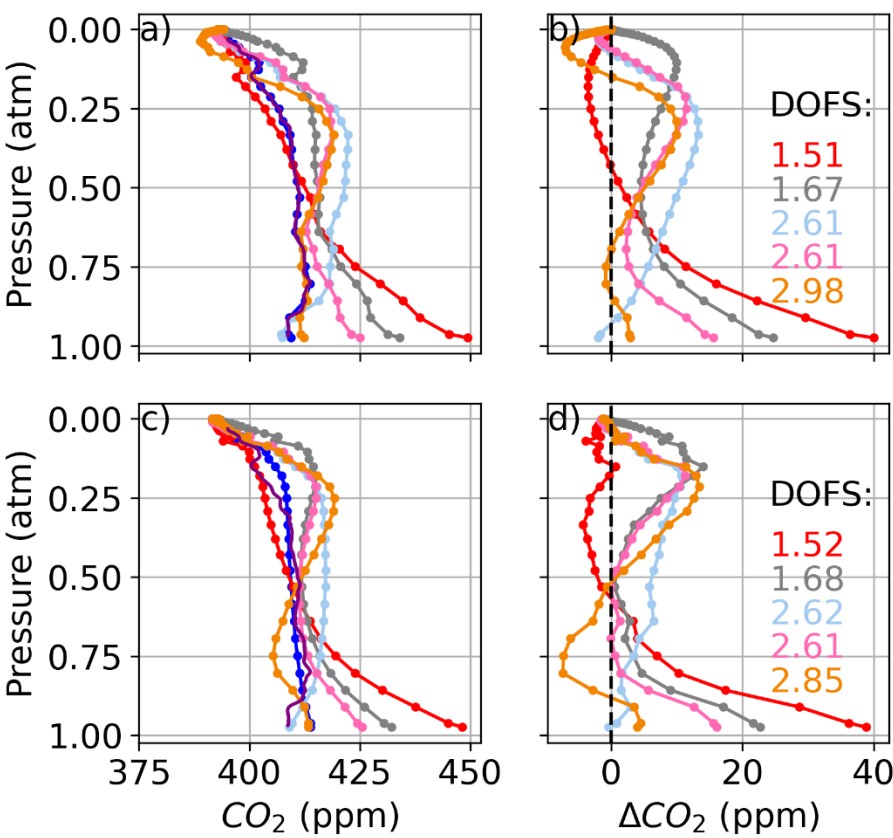

**Figure 9: Same as Fig. 8 but for spectra measured on 11 April 2017 at 28–39° solar zenith angle.**

455    Figure 10 shows the difference between the GGG2020 a priori temperature profile, used in Fig. 8(c) and 9(c), and the radiosonde temperature profile used in Fig. 8(a) and 9(a). In both cases, we replace the a priori surface temperature with the measured surface temperature. On 14 January 2012, the radiosonde temperature profile is about 1 °C higher than the GGG2020 a priori profile at pressures < 0.6 atm. The shape of the Strong window $CO_2$ profile deviations in Fig. 8(c) is consistent with the sensitivity tests using synthetic spectra in Sect. 3.1.3. In Fig. 5(a), a +5°C offset below 5 km results in +600 ppm $CO_2$ error

460    at ~0.9 atm, while in Fig. 9, a -1°C offset in the lower troposphere leads to a -50 ppm error at ~0.9 atm. The deviations are smoother in Fig. 8 and 9 than in Fig. 5 because the SNR of real spectra is between 200 and 500 instead of 1000, and because of the smoothing effect of the off-diagonal elements of the a priori covariance used in this section. The off-diagonal elements of the a priori covariance introduce inter-layer correlations that reduce large differences between levels over a given length scale (see Sect. 2.2). Retrievals on real spectra after applying a +5°C offset to the radiosonde temperature profile below 5 km

465    lead to a +100 ppm offset at ~0.9 atm. The $CO_2$ profiles in Fig. 9(c) differ less with those in Fig. 9(a) than do the profiles in





Fig. 8(a) and 8(c). In Fig. 10, the difference between the GGG2020 and radiosonde temperature profile on 11 April 2017 is ~3°C for the first two levels above the surface, but the average difference between 0.85 and 0.6 atm is -0.15°C compared to -1.05°C on 14 January 2012.

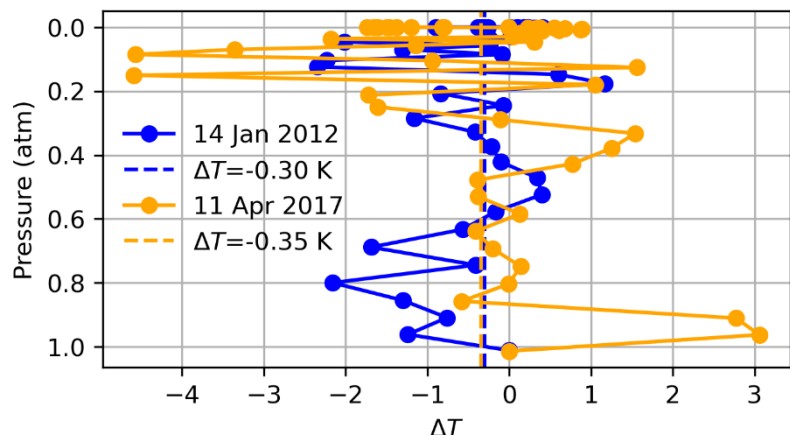

**Figure 10: Temperature profile difference for the GGG2020 a priori minus radiosonde on 14 January 2012 and 11 April 2017. The radiosonde profile is included in the a priori used in panel (a) of Figs. 8 and 9, and the GGG2020 a priori is used in panel (b) of Figs. 8 and 9. In situ temperature measurements are used for both cases at the surface. The dashed line marks the average difference, with the value indicated in the legend.**

In aircraft profiles over Lamont between 2008 and 2018 from NOAA's ObsPack, the steepest vertical gradients in $CO_2$ profiles are ~5 ppm/km between the surface and ~3 km. In its current state, $CO_2$ profile retrieval with GFIT2 cannot distinguish these vertical variations from $CO_2$ deviations caused by errors in the forward model, even with very accurate a priori meteorology. Typical errors in the a priori temperature profiles will prevent operational use of $CO_2$ profile retrieval without a scheme for retrieving or correcting the temperature profiles.

**3.2.2 Information content and averaging kernel**

Table 5 presents the average values of the Shannon information content, $H$, and of the $CO_2$ profile DOFS, from all profile retrievals performed on Lamont spectra when using the GGG2020 a priori profiles. It also includes the Ratio of Residuals (RR) of the spectral fits (see Appendix B, Eq. B10), which represents the residuals of the profile retrievals as a fraction of the residuals of the scaling retrievals. The same quantities are plotted in Fig. 11 for each spectrum. The RR is always smaller than 1 because the profile retrieval has more freedom to adjust the calculated spectrum and so can never produce larger residuals than scaling retrievals.





**Table 5: Shannon information content (H), degrees of freedom for signal (DOFS) for the CO2 profile, and Ratio of Residuals (RR)**
**averaged over all 492 profile retrievals from near-infrared TCCON spectra measured at Lamont. The standard deviation is also**
**shown.**

| Window name | H | DOFS | RR |
|---|---|---|---|
| TCCON1 | 5.4±0.6 | 2.7±0.2 | 0.988±0.014 |
| TCCON2 | 5.4±0.6 | 2.7±0.2 | 0.992±0.009 |
| Weak1 | 2.3±0.7 | 1.7±0.3 | 0.996±0.002 |
| Weak2 | 2.5±0.9 | 1.8±0.4 | 0.994±0.008 |
| Strong | 6.8±1.0 | 3.0±0.4 | 0.957±0.038 |

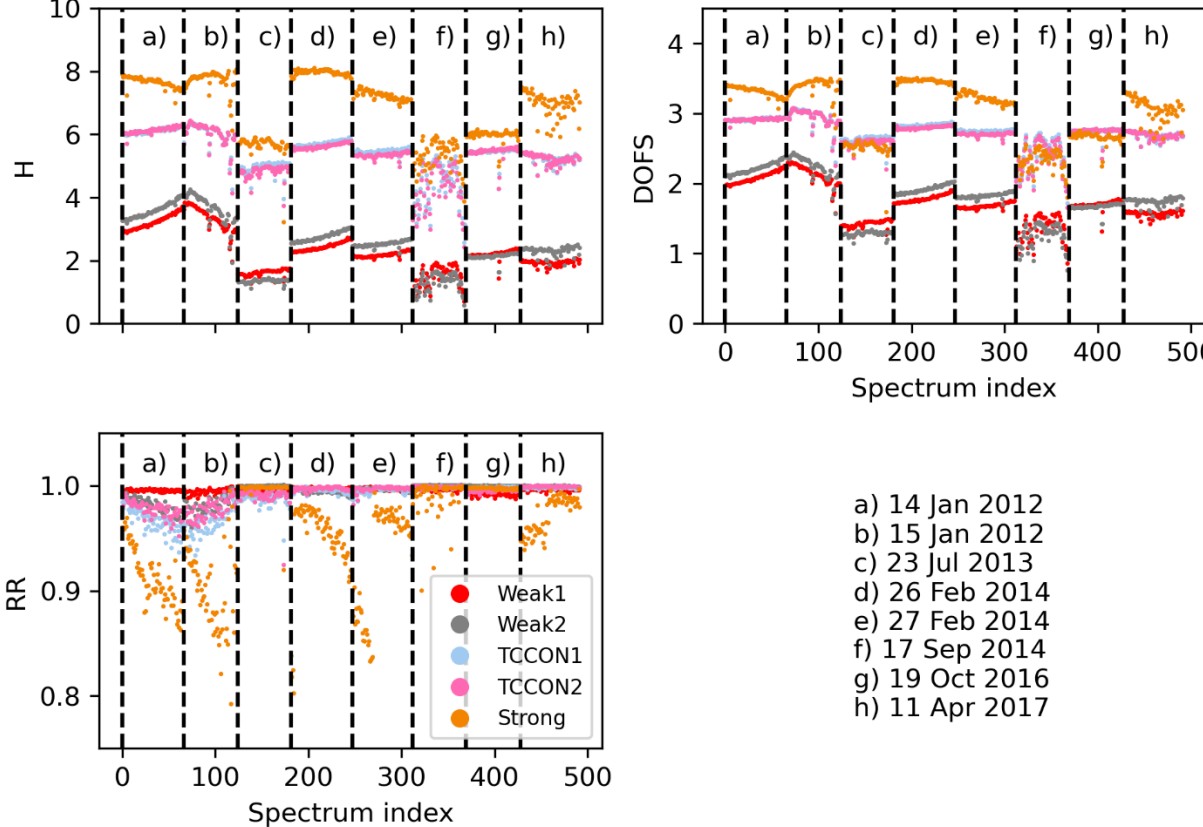

**Figure 11: Shannon information content (top left), degrees of freedom for signal for the CO2 profile (top right), and ratio of residuals**
**(bottom left) for all Lamont spectra coincident within ±1 h of the AirCore last sampling time for AirCores launched on the dates**
**indicated in the bottom right. Each new date is marked by a vertical dashed line.**





Figure 12 shows the sums of the rows of the partial column averaging kernel matrix over different altitude ranges. The sum from 0 to 70 km is the total column averaging kernel (see Appendix C). The total column averaging kernel is close to 1 at all

levels in all windows, indicating good sensitivity to changes in the $CO_2$ total column. The partial column kernels show that most of this sensitivity comes from altitudes below 15 km. That the total column averaging kernel is close to 1 at all levels is not inconsistent with the large deviations we observe in the retrieved $CO_2$ profiles. If the total column averaging kernel is exactly one at each level, adding $N$ molecules of $CO_2$ anywhere in the atmosphere will lead to $N$ more molecules in the retrieved total column. However, in the presence of a priori temperature errors, for example, the retrieved value can be biased. The

averaging kernel indicates that without the effect of these errors, the $CO_2$ profile retrieval would have excellent sensitivity to $CO_2$ and would be able to provide information about $CO_2$ in two distinct layers. Here, the vertical representation is not a concern. Using 51 vertical levels only affects the speed of the retrieval. The retrieved profiles can then be reduced to a number of partial columns corresponding to the DOFS. This was not done here because it is evident that large deviations due to temperature errors could easily bias the resulting partial columns. The reduction into a subset of layers also requires an arbitrary

choice: in Fig. 12 the altitude ranges were set such that the DOFS of the first two partial columns would be roughly close to 1 in each window. We could also have chosen two regions with approximately equal DOFS from 0–7 km and 7–70 km. The partial column averaging kernels overlap with each other, so the partial columns are not completely uncorrelated even if their respective DOFS are higher than 1. The DOFS are not exactly independent pieces of information, as it is impossible to obtain independent partial column amounts from direct sun measurements on the ground (see Appendix C), but an arbitrary criterion

can be defined to identify distinct layers, for example if the peaks in their partial column averaging kernels are separated by a given fraction of their widths in altitude. Additional analysis of the vertical sensitivity of the retrieval is presented in Appendix D.



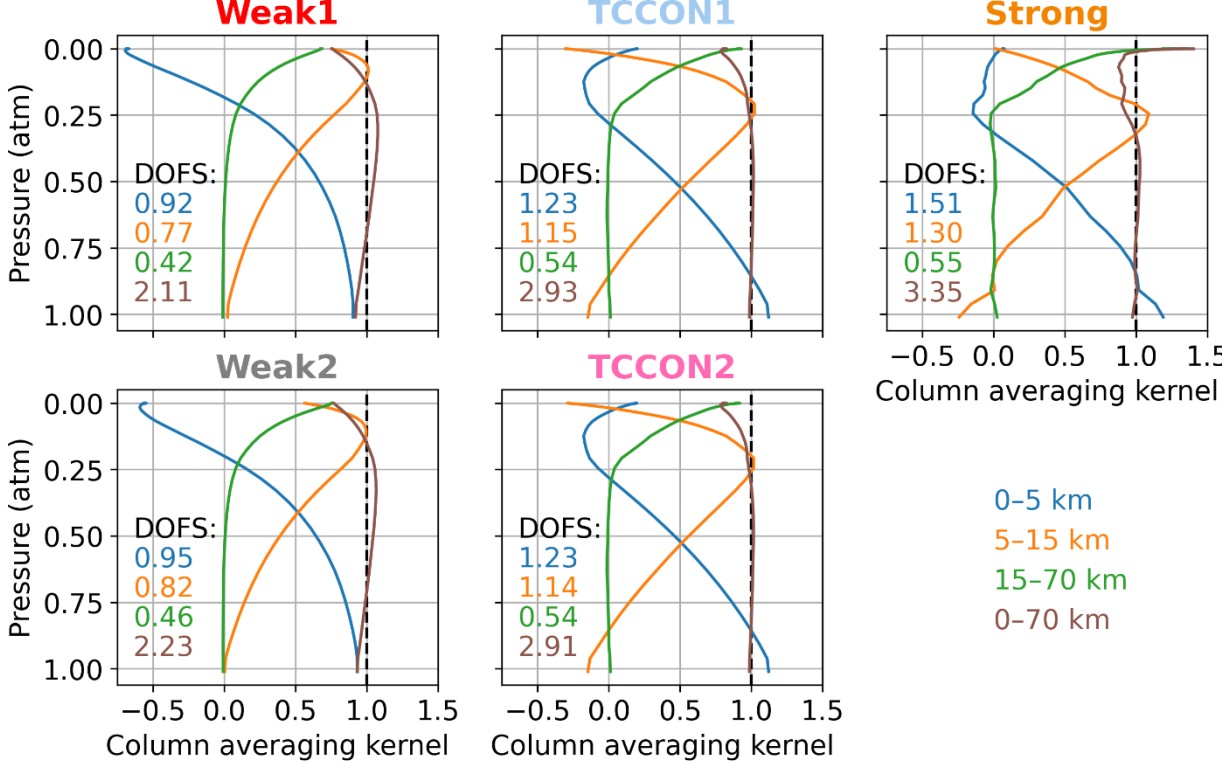

**Figure 12: Sum of the rows of the partial column averaging kernel matrix over different altitude ranges as indicated by the legend,** for each of the five $CO_2$ windows. The sum between 0–70 km is the total column averaging kernel. The numbers in each panel are the DOFS corresponding to each of the altitude ranges.

### 3.2.3 XCO2

The $XCO_2$ derived from profile and scaling retrievals using the GGG2020 a priori was compared to $XCO_2$ derived from the $CO_2$ profile built from the AirCore $CO_2$ profile, in situ surface measurements of $CO_2$, and the GGG2020 a priori $CO_2$ above the maximum altitude sampled by the AirCore. The results are shown in Fig. 13 for the eight days we have AirCore profiles that are coincident with measurements at the Lamont TCCON station. Despite the large deviations observed in retrieved profiles, the $XCO_2$ derived from profile retrievals compares well to the AirCore $XCO_2$, but it does not present a clear improvement over the $XCO_2$ derived from the scaling retrievals. The effect of temperature errors on $XCO_2$ derived from scaling and profile retrievals is relatively small because the spectral windows utilize the entire (fundamental) band. Across a wide window, the residuals due to temperature errors show alternating positive and negative residuals, because of the different temperature sensitivities of absorption lines. Collectively, these lines have a small net temperature sensitivity. The scaling retrieval, which can only add or remove $CO_2$ at all levels simultaneously, is limited in its ability to fit out such residuals across a wide window by adjusting the $CO_2$ scale factor. For profile retrievals, although large deviations are observed in the retrieved profile, they compensate each other when deriving the total column. These deviations compensate due to the wide windows





including a range of spectral lines with different temperature sensitivities. If a narrow window over only a few lines were used

instead, we would expect more localized errors in the retrieved $CO_2$ profiles, and total columns sensitive to temperature errors.

**Figure 13: XCO2 derived from scaling (dashed lines and squares) and profile (solid lines and circles) retrievals for each $CO_2$ window when using the GGG2020 a priori, compared to $XCO_2$ derived from smoothed AirCore profiles (see Appendix C). The black dotted**
**line marks the 1-to-1 line. When comparing with scaling retrievals, the AirCore profile is smoothed using the total column averaging kernel of the scaling retrieval, and when comparing to profile retrievals the AirCore profile is smoothed using the averaging kernel matrix of the profile retrieval. The legend indicates the slopes and squared Pearson correlation coefficients of fits to lines passing through the origin, assuming that in the absence of $CO_2$ the retrieval would return a $CO_2$ value of zero.**





### 545   3.2.4 Real spectra: discussion

Profile retrievals that use real spectra and an a priori profile built from coincident in situ measurements show $CO_2$ profile deviations up to 40–50 ppm. Even when the errors due to the a priori meteorology are minimized, deviations from the truth due to instrument misalignment, radiative transfer, sun-tracker pointing, or uncertainties in line parameters are larger than typical vertical variations of $CO_2$.

When performing retrievals on the same spectra but replacing the AirCore a priori profile with a standard a priori profile, small errors in the a priori temperature profile cause large deviations in the retrieved $CO_2$ profile. Despite the large deviations in the retrieved profiles, the retrieval still shows high sensitivity to $XCO_2$ but does not present a clear improvement over $XCO_2$ obtained from scaling retrievals. Introducing a temperature retrieval or correction, as well as the ability to model an imperfect instrument line shape, is the best avenue to improve the $CO_2$ profile retrieval results. Appendix E presents an attempt at 555   applying empirical corrections to reduce the effect of systematic imperfections in the forward model.

### 4.   Summary and conclusions

In this study we investigated the use of $CO_2$ profile retrievals from near-infrared solar absorption spectra measured by TCCON. The performance of $CO_2$ profile retrieval was reassessed after improvements were implemented in the forward model of GGG. Retrievals were performed using five $CO_2$ windows with significantly different optical opacities.


We first use retrievals on synthetic spectra to check the self-consistency. Typical errors in the a priori $H_2O$ profile, which is retrieved with a scaling retrieval, caused limited deviations from the truth in the $CO_2$ profile, within 5–10 ppm in the Strong window, and within 2 ppm in the other windows. Perturbing the $CO_2$ air- and self-broadened Lorentz half-width coefficients and their temperature dependence to within their estimated uncertainties led to $CO_2$ deviations from the truth of less than 5 565   ppm. The implementation of a non-Voigt line shape is a significant improvement to $CO_2$ profile retrievals; errors in spectroscopic parameters are no longer the leading source of uncertainty in retrieved profiles. We observed deviations from the truth of up to 100 ppm in profiles retrieved with typical temperature errors. The temperature profile is an important retrieval input, but is not retrieved, thus spectral residuals caused by errors in the a priori temperature profile are free to be suppressed by adjustments to the $CO_2$ scale factors. The implementation of a temperature profile retrieval, or correction, is critical to 570   improve $CO_2$ profile retrieval results. In GGG2020, 3-hourly a priori temperature profiles are used, but temperatures can still vary by several degrees between 3-hourly profiles and can still be wrong even without any time mismatch. Temperature could be retrieved from $CO_2$ windows and from windows with temperature-sensitive water vapour absorption lines.

We then perform retrievals with atmospheric TCCON spectra collected at the Lamont site, which were coincident with AirCore 575   profiles, including radiosonde profiles of temperature and relative humidity; these were considered as the true state of the atmosphere. When running retrievals with the truth as the a priori, the deviations due to errors in the a priori meteorology are





minimized and the resulting deviations are caused by instrument misalignment, errors in spectroscopy, or sun tracker pointing. We observed $CO_2$ deviations of up to 40 ppm in that case. Even with ideal knowledge of the a priori meteorology, the $CO_2$ deviations are larger than the largest expected vertical $CO_2$ variations and no useful profile information can be inferred from

the profile retrieval. Stricter alignment requirements, which can be challenging to achieve in practice, or the ability to model an imperfect instrument line shape are needed to improve profile retrieval results. The sensitivity study of Sect. 3.1 could then be extended to assess the effect of specific misalignments on the retrieved profiles.

In these retrievals, we used a full a priori covariance matrix, with off-diagonal elements, based on comparisons between the

GGG2020 a priori and aircraft vertical profiles from NOAA's ObsPack over the Lamont TCCON site. Before tuning the a priori covariance and considering stronger regularisations, it must be shown that $CO_2$ deviations caused by typical errors in the a priori meteorology are smaller than typical variability in real $CO_2$ profiles. Because it is more computationally expensive, and because it requires stronger constraints on the a priori statistics than scaling retrievals, a profile retrieval must present clear advantages over a scaling retrieval to justify its operational use. And with each new improvement to the $CO_2$ a priori profiles,

requirements for profile retrieval to be better than scaling retrieval become more stringent.

A method to combine the profiles obtained from sequential retrievals in different spectral windows still needs to be developed. Alternatively, the ability to perform simultaneous retrievals using multiple spectral windows could be implemented in GFIT2.

**Appendices**

**Appendix A: Vertical columns**

The vertical grid for the retrievals presented in this study has 51 levels from 0 to 70 km, with spacing increasing with altitude and following:

$$z_i = i \times (0.4 + 0.02 \times i) \qquad (A1)$$

where $z_i$ is the altitude in kilometers of the $i^{th}$ level. Each level is associated with an effective vertical path distance $vp$:

$$vp_i \approx 0.4 + 0.04 \times i \qquad (A2)$$

The total column of air in molecules per square meter can be obtained as:

$$column_{Air} = \sum_{i=1}^{N} vp_i \times d_i \qquad (A3)$$

where $d$ is the air number density in molecules of air per cubic meter. $N$ is the number of atmospheric levels. If the prescribed grid contains layers below the altitude of the site considered, their effective vertical path will be 0. The layer containing the

site altitude will be truncated. The total column of $CO_2$ is:





$$column\,n_{CO2} = \sum_{i=1}^{N} sf_i \times vmr_i \times d_i \times vp_i \qquad (A4)$$

where *sf* is the retrieved scaling factor and *vmr* is the a priori $CO_2$ wet mole fraction (molecules of $CO_2$ per molecules of air).
In the forward model, the retrieval grid is not vertical, but along the slant path from the instrument towards the sun, the scaling
factors retrieved for the slant layers are used with the corresponding vertical layers to compute the vertical column. The a priori

profiles used by GFIT are built on the prescribed altitude grid directly above the site. This should contribute to an unknown
error, largest at high solar zenith angles when the projection of the sun ray on the ground can reach a few hundred kilometers;
in that case the a priori slant profiles of temperature and $H_2O$ could be significantly different from the vertical profile directly
above the instrument.

The column-averaged dry-air mole fraction of $CO_2$ ($XCO_2$) is the ratio of the column of $CO_2$ to the column of dry air, where
the column of dry air is expressed as the column of $O_2$ divided by 0.2095 (Wunch et al., 2011b):

$$XCO_2 = 0.2095 \times \frac{\overline{column\,n_{CO2}}}{column\,n_{O2}} \qquad (A5)$$

where the $O_2$ column is retrieved from a spectral window centered at 7885 $cm^{-1}$. For the official TCCON products, $\overline{column\,n_{CO2}}$
is a weighted average of the columns retrieved from the TCCON1 and TCCON2 windows.


**Appendix B: GFIT2 algorithm**

To find the state vector with maximum a posteriori probability given a measurement, the cost function J is minimized:

$$J = \left(y - f(x)\right)^T S_y^{-1}\left(y - f(x)\right) + (x_a - x)^T R(x_a - x) \qquad (B1)$$

by iteratively solving for the state update $\Delta x$ in the least square problem:

$$\left(K_i^T S_y^{-1} K_i + R + \gamma D\right) \Delta x = K_i^T S_y^{-1}\left(y - f(x_i)\right) + R(x_a - x_i), \qquad (B2)$$

Here, $y$ is the measured transmittance spectrum, f is the forward model that computes a transmittance spectrum from the state
vector $x$, $S_y$ is the measurement covariance matrix, $x_a$ is the a priori state vector, and the regularisation matrix $R$ is taken to be
the inverse of the a priori covariance matrix $S_a$. $K$ is the Jacobian matrix, each column of $K$ contains the derivative of the
spectrum with respect to an element of the state vector, $K = \frac{\partial f(x)}{\partial x}$. The Levenberg–Marquardt parameter $\gamma$ is applied to a

scaling matrix $D$, which is also taken to be $S_a^{-1}$. The Levenberg–Marquardt parameter affects the size of the state update so
that smaller steps may be taken when the linearization of the forward model is not satisfactory.

The expected $\chi^2$ of the maximum a posteriori probability solution should be:

$$\chi^2(\hat{x} - x) = (\hat{x} - x)^T\left(K^T S_y^{-1} K + S_a^{-1}\right)(\hat{x} - x) \approx n \qquad (B3)$$





where $n$ is the number of state vector elements. A solution is accepted when the ratio of the squared state update to the estimated variance is a negligible fraction of the expected $\chi^2$:

$$\Delta x \left(\mathbf{K}^T \mathbf{S}_y^{-1}(\boldsymbol{y} - f(\boldsymbol{x}_i)) + \mathbf{S}_a^{-1}(\boldsymbol{x}_a - \boldsymbol{x}_i)\right) \ll n. \tag{B4}$$

In an algorithm, "$\ll n$" must use a specific limit, and in GFIT2, "$< n/10$" was used. If the inequality check is made with a parameter that is too large, like "$<n$", the algorithm may take fewer iterations to converge, but will take the same steps at each iteration, often leading to a retrieved profile closer to the a priori. The inequality check should be done with a small enough fraction of $n$ that reducing it further does not significantly affect the solution.

If convergence is not reached in the i$^{th}$ iteration, an algorithm determines if the Levenberg–Marquardt parameter needs to be adjusted for the next iteration (Fletcher, 1971). Three different cost functions are used:

$$J_{old} = \left(\boldsymbol{y} - f(\boldsymbol{x}_i)\right)^T \mathbf{S}_y^{-1}\left(\boldsymbol{y} - f(\boldsymbol{x}_i)\right) + (\boldsymbol{x}_a - \boldsymbol{x}_i)^T \mathbf{S}_a^{-1}(\boldsymbol{x}_a - \boldsymbol{x}_i) \tag{B5}$$

$$J_{new} = \left(\boldsymbol{y} - f(\boldsymbol{x}_i + \Delta\boldsymbol{x})\right)^T \mathbf{S}_y^{-1}\left(\boldsymbol{y} - f(\boldsymbol{x}_i + \Delta\boldsymbol{x})\right) + (\boldsymbol{x}_a - \boldsymbol{x}_i - \Delta\boldsymbol{x})^T \mathbf{S}_a^{-1}(\boldsymbol{x}_a - \boldsymbol{x}_i - \Delta\boldsymbol{x}) \tag{B6}$$

$$J_{pred} = \left(\boldsymbol{y} - f(\boldsymbol{x}_i) - \mathbf{K}\Delta\boldsymbol{x}\right)^T \mathbf{S}_y^{-1}\left(\boldsymbol{y} - f(\boldsymbol{x}_i) - \mathbf{K}\Delta\boldsymbol{x}\right) + (\boldsymbol{x}_a - \boldsymbol{x}_i - \Delta\boldsymbol{x})^T \mathbf{S}_a^{-1}(\boldsymbol{x}_a - \boldsymbol{x}_i - \Delta\boldsymbol{x}) \tag{B7}$$

where $J_{old}$ is the cost function using the state vector at the beginning of the i$^{th}$ iteration, $J_{new}$ is the cost function using the updated state vector at the end of the i$^{th}$ iteration, and $J_{pred}$ is the cost function using the state vector update and the linear approximation:

$$f(\boldsymbol{x} + \Delta\boldsymbol{x}) \cong f(\boldsymbol{x}) + \mathbf{K}\Delta\boldsymbol{x}. \tag{B8}$$

The ratio r is then evaluated:

$$r = \frac{J_{new} - J_{old}}{J_{pred} - J_{old}}. \tag{B9}$$

This is the relative change in the cost function produced by a state vector update when using the forward model and a linear approximation of the forward model. The Levenberg–Marquardt parameter is then adjusted as follows:

- r > 0.75: the linearization of the forward model is satisfactory and $\gamma$ is reduced to allow larger steps
  - $\gamma = \frac{\gamma}{2}$
- r ≥ 0.25: intermediate case, make no change to $\gamma$ and reset the number of consecutive divergences
  - ndiv = 0
- r < 0.25: the linearization of the forward model is not satisfactory, increment the number of consecutive divergences, $\gamma$ is increased to take smaller steps.
  - ndiv=ndiv+1
  - if , $\gamma = 0$ then , $\gamma$=1
  - if , $\gamma > 0$ then , $\gamma = 10\gamma$

If ndiv reaches some specified maximum number, there will not be another iteration. When r < 0.25, it means that the linearization of the forward model is not good enough. In GFIT2, this was not allowed to happen more than twice in a row.





Increasing $\gamma$ will lead to a smaller step for the state vector update, increasing the chance that the linearization of the forward model at the next step will be better and $r \geq 0.25$.

In GFIT2 $r > 0.75$ in most cases, and if $\gamma$ is not initially set to 0 it will tend towards zero until the convergence criterion is met, thus the initial value of $\gamma$ was set to 0. However, the increase of the parameter is often triggered when fitting noisier spectra and can give the algorithm a chance to converge when it would otherwise need more iterations or fail without $\gamma$.

After the last iteration, the goodness of the retrieval is checked by evaluating the reduced $\chi^2$ of the spectral residuals.

$$\chi_{red}^2(\boldsymbol{y} - f(\boldsymbol{x})) = \frac{1}{N}\sum_{i=1}^{N}\left(\frac{\boldsymbol{y}_i - f(\boldsymbol{x})_i}{y_{noise}}\right)^2 \qquad (B10)$$

where $y_{noise}$ is the measurement uncertainty, and N the number of spectral points. Profile retrieval from real spectra are presented in Sect. 3 where the root mean square of the residuals from a scaling retrieval is used as $y_{noise}$. In that case Eq. B10 is the average Ratio of Residuals (RR) between the profile and scaling retrieval.

The retrieval covariance matrix is:

$$\widehat{\boldsymbol{S}} = \left(\boldsymbol{K}^T \boldsymbol{S}_y^{-1}\boldsymbol{K} + \boldsymbol{S}_a^{-1}\right)^{-1}. \qquad (B11)$$

The square root of its diagonal elements is used as the uncertainty on the retrieved scaling factors.

**Appendix C: Averaging kernel**

The state vectors of GFIT and GFIT2 contain scaling factors to be applied to a priori mole fractions. The averaging kernel matrix is:

$$\mathbf{A} = \left(\boldsymbol{K}^T \boldsymbol{S}_y^{-1}\boldsymbol{K} + \boldsymbol{S}_a^{-1}\right)^{-1}\boldsymbol{K}^T \boldsymbol{S}_y^{-1}\mathbf{K}. \qquad (C1)$$

It is a change in the retrieved state for a change in the state vector elements.

$$(\boldsymbol{A}_{SF})_{i,j} = \frac{\delta \widehat{\boldsymbol{x}}_i}{\delta \boldsymbol{x}_j} \qquad (C2)$$

Even though the averaging kernel is dimensionless, its units can be written as e.g., "ppm per ppm" to indicate that it is the change at a given level for a change at a different level.

To obtain the averaging kernel in ppm per ppm:

$$(\mathbf{A}_{VMR})_{i,j} = (\mathbf{A}_{SF})_{i,j}\frac{vmr_i}{vmr_j} \qquad (C3)$$

where $vmr$ is the a priori mole fraction at the i$^{th}$ and j$^{th}$ levels and the partial column averaging kernel matrix in molecules.cm$^{-2}$ per molecules.cm$^{-2}$ is:



$$(A_{col})_{i,j} = (A_{SF})_{i,j} \frac{vmr_i \times d_i \times sp_i}{vmr_j \times d_j \times sp_j} \tag{C4}$$

Where $sp$ are the widths of the slant layers along the sun ray that correspond to the altitude levels of the prescribed vertical grid. The total column averaging kernel vector can be obtained from the partial column averaging kernel matrix:

$$a_j = \sum_{i=1}^{nlev} (A_{col})_{i,j}. \tag{C5}$$

It represents the change in the total column (molecules.cm$^{-2}$) caused by a change in the partial column of the j$^{th}$ layer. It should ideally be equal to 1 at each level, meaning that adding N target molecules anywhere in the atmosphere will lead to N more molecules in the retrieved total column.

The averaging kernel matrix would ideally be an identity matrix, meaning that adding N molecules in the j$^{th}$ layer would lead

to N more molecules retrieved in that layer. However, adding N molecules in the j$^{th}$ layer will lead to an increase in the width of $CO_2$ absorption lines of a spectrum observed from the ground. As illustrated for $CO_2$ in Fig. 2, each wavenumber is affected by the $CO_2$ concentration over a range of altitudes, because the spectrum observed on the ground is the product of all the spectra that would be observed at each altitude. Even if that change in line widths was the only change in the spectrum and could be fitted perfectly, it would be impossible to exactly attribute that change to a specific level. Although the total column

averaging kernel could be exactly 1 at each level, the averaging kernel matrix can never be exactly the identity matrix for direct sun measurements from the ground.

The column averaging kernel matrix can be used to degrade higher resolution profiles before comparing them to retrieved profiles (Rodgers and Connor, 2003).

$$c_s = A_{col}(c - c_a) + c_a \tag{C6}$$

where $c_s$ is the smoothed partial column profile, $c$ is the retrieved partial column profile, and $c_a$ is the a priori partial column profile. Or using the total column averaging kernel:

$$c_s^{tot} = c_a^{tot} + a^T(c - c_a) \tag{C7}$$

where $a^T$ is the transpose of $a$, and the "tot" superscript indicates a total column:

$$c_a^{tot} = \sum_{i=1}^{nlev} c_{a_i} \tag{C8}$$

**Appendix D: Information content and error analysis**

The singular value decomposition of the $CO_2$ Jacobian matrix can provide information on the relative precision with which different vertical patterns are measured. The Jacobian matrix $K$ is decomposed into:

$$\mathbf{K}(nmp, nlev) = \mathbf{U}(nmp, nlev)\mathbf{L}(nlev, nlev)\mathbf{V}^T(nlev, nlev) \tag{D1}$$





where *nmp* is the number of measured spectral points, *nlev* is the number of atmospheric levels, **U** is the matrix of left singular vectors, $L$ is the diagonal matrix of singular values, and $\mathbf{V}^T$ is the transpose of the matrix of right singular vectors. The right singular vectors of **K** associated with the eight largest singular values are shown in Figs. D1 to D5 for each $CO_2$ band on 14 January 2012. The right singular vectors represent independently measured vertical patterns with a precision indicated by their corresponding singular values shown above each panel. The singular values are also shown as a fraction of the largest singular value in parenthesis. The singular vectors all show an increasing number of oscillations with decreasing singular value. In each

window, the first singular vector is close to a uniform weighting at all altitudes and has 3 to 10 times more sensitivity than the second pattern. The singular vector in panel (d) has a structure like that of the $CO_2$ profile deviations observed in the sensitivity tests of Sect. 3.1.

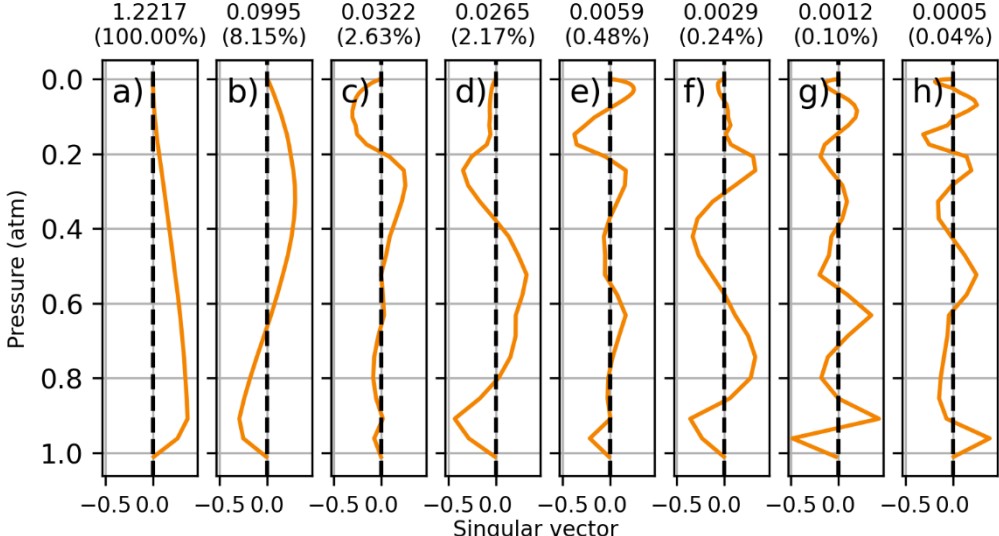

**Figure D1: Right singular vector of the Jacobian associated with the eight largest singular values for profile retrievals from the Strong $CO_2$ window on 14 January 2012. The singular values are shown above each panel, and the singular value normalized to the largest singular value is shown in parenthesis.**





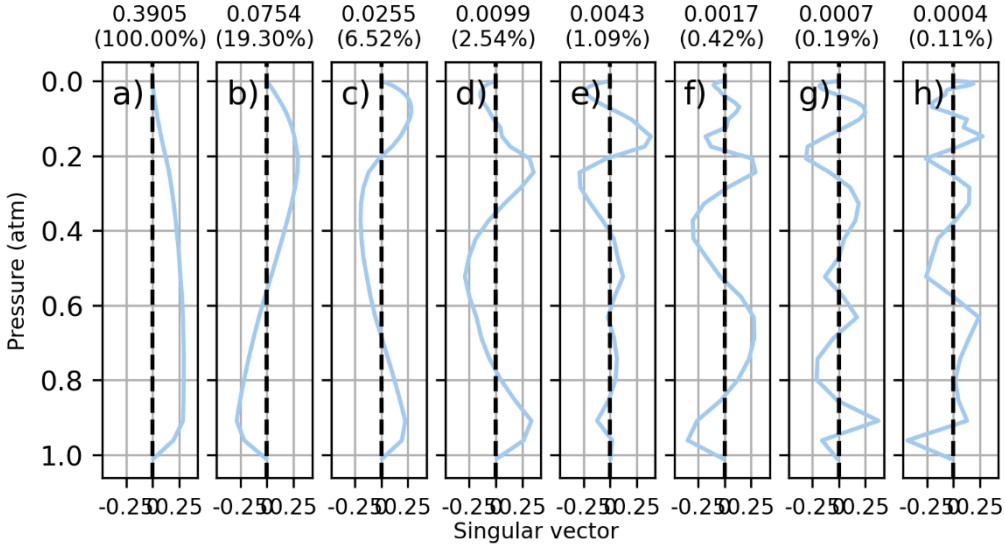

**Figure D2: Same as Fig. D2 but for the TCCON1 window.**

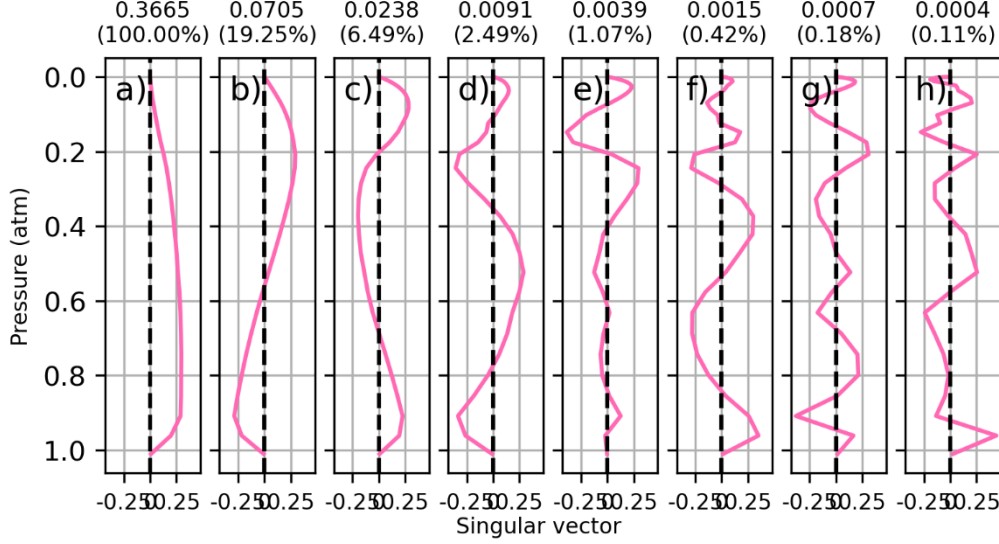


**Figure D3: Same as Fig. D1 but for the TCCON2 window.**





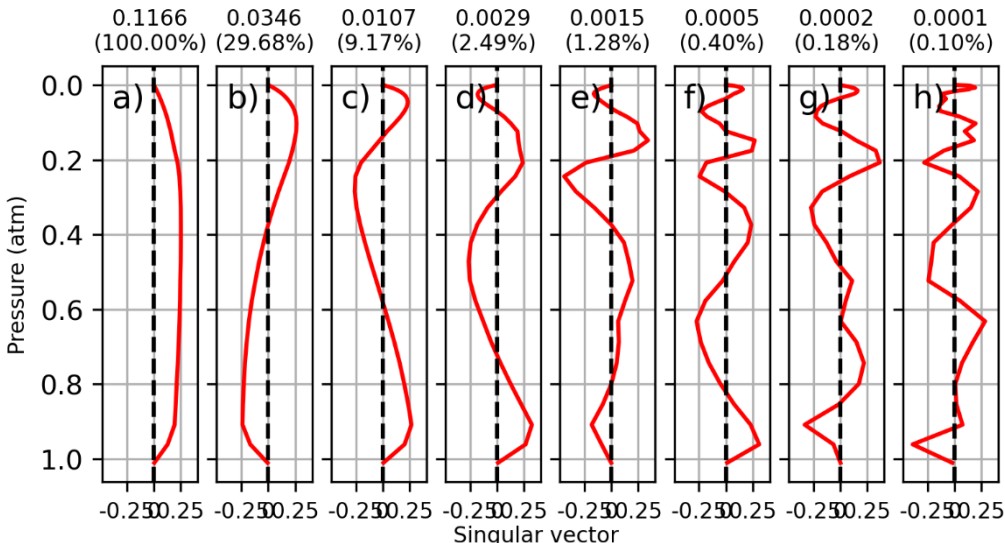

**Figure D4: Same as Fig. D1 but for the Weak1 window.**

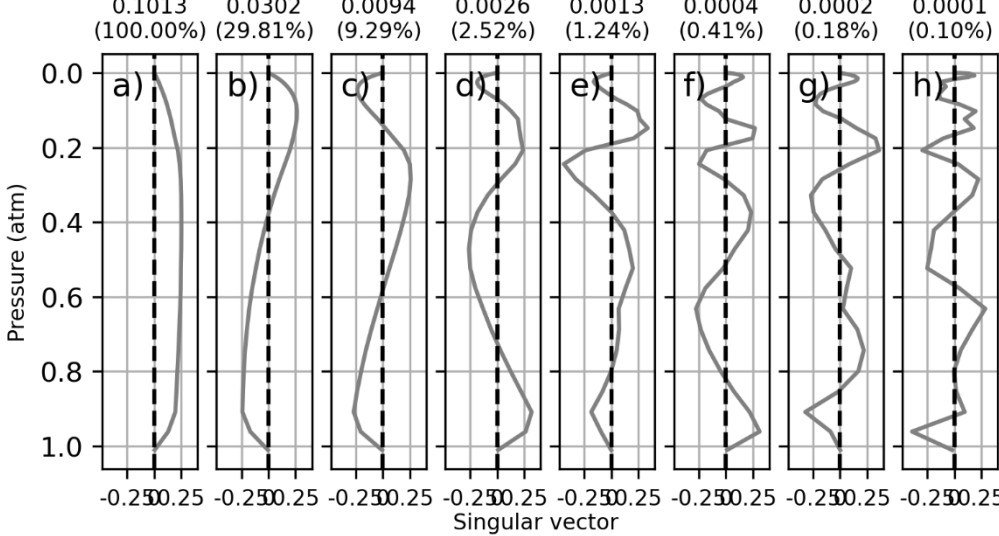

**Figure D5: Same as Fig. D1 but for the Weak2 window.**

The retrieval covariance matrix $\hat{\mathbf{S}}$ can be expressed as a sum of the null space covariance $\mathbf{S}_N$ and the measurement noise covariance $\mathbf{S}_m$ (Rodgers, 1990):

$$\mathbf{S}_N = (\mathbf{S}_a^{-1} + \mathbf{K}^T\mathbf{S}_e^{-1}\mathbf{K})^{-1}\mathbf{S}_a^{-1}(\mathbf{S}_a^{-1} + \mathbf{K}^T\mathbf{S}_e^{-1}\mathbf{K})^{-1} \tag{D2}$$

$$\mathbf{S}_m = (\mathbf{S}_a^{-1} + \mathbf{K}^T\mathbf{S}_e^{-1}\mathbf{K})^{-1}\mathbf{K}^T\mathbf{S}_y^{-1}\mathbf{K}(\mathbf{S}_a^{-1} + \mathbf{K}^T\mathbf{S}_e^{-1}\mathbf{K})^{-1} \tag{D3}$$



The error patterns of these matrices hold information on vertical structures in the $CO_2$ profiles that the retrieval cannot resolve, due to the smoothing effect of the a priori covariance matrix $\boldsymbol{S}_a$ in the case of $\boldsymbol{S}_N$, and due to the effect of measurement noise in the case of $\boldsymbol{S}_m$, as the measurement error covariance matrix $\boldsymbol{S}_y$ only represents random errors in the measured radiances. The error patterns of a matrix are defined as its eigenvectors multiplied by the square root of their corresponding eigenvalue.

The error patterns of $\boldsymbol{S}_N$ associated with the four largest eigenvalues are shown in Fig. D6, and those of $\boldsymbol{S}_m$ are shown in Fig. D7. In both cases, the largest error pattern peaks at the surface and falls to 0 at ~0.9 atm; these peaks in the error patterns correspond to a minimum in the singular vectors of the $CO_2$ Jacobian. The large errors in the retrieved $CO_2$ profiles are explained by the larger a priori uncertainty in the lower troposphere, and by the relatively larger effect of errors at wavenumbers strongly weighted at low altitudes. This is because "sensitivity" is determined by the Jacobian; the retrieval will simply

preferentially adjust $CO_2$ at levels where a given change in $CO_2$ causes a larger change in radiance. At pressures larger than ~0.9 atm, the error patterns of $\boldsymbol{S}_N$ represent vertical scales that cannot be resolved in the retrieval, with a vertical scale of 0.3 atm or less.

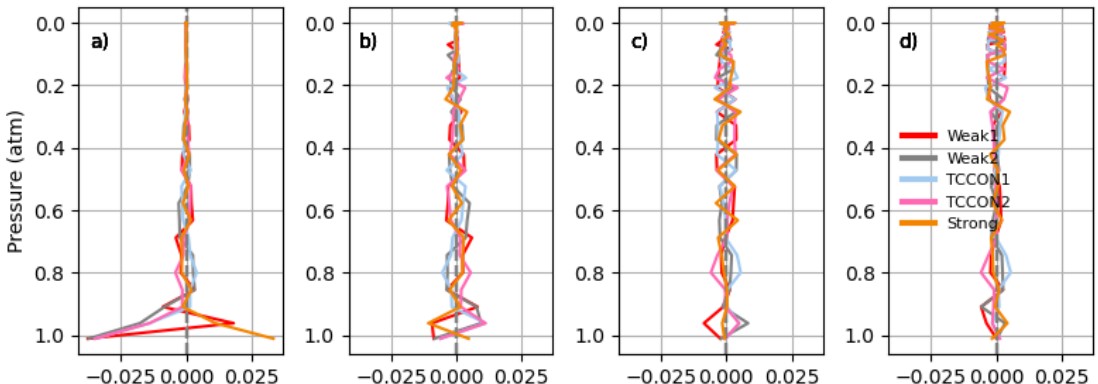

**Figure D6: The four largest error patterns of the null space covariance matrix for a Lamont spectrum measured on 14 January**
**2012.**

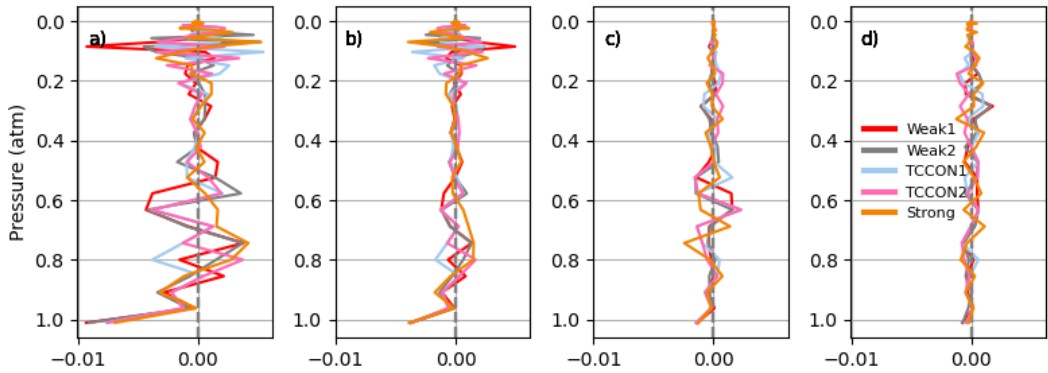

**Figure D7: Same as Fig. D6 but for the measurement noise covariance matrix.**



The uncertainty on the retrieved $CO_2$ profile is taken to be the square root of the diagonal elements of $\hat{S}$ even though the retrieval covariance is not diagonal. It is presented in Fig. D8 as a percentage of the a priori uncertainties. The retrieval error

is always smaller than the a priori covariance by construction in optimal estimation, so this alone gives no indication of a successful retrieval. But the retrieval is more sensitive to altitudes where the retrieval uncertainty is a smaller fraction of the a priori uncertainty. The error from the diagonal of $\mathbf{S}_N$ and $\mathbf{S}_m$ is also shown. In addition to $\mathbf{S}_N$, the smoothing contribution from state vector elements other than $CO_2$ scale factors is shown as $\mathbf{S}_i$, the interference error covariance (Rodgers and Connor, 2003):

$$\mathbf{S}_i = \boldsymbol{A}_{xe}\boldsymbol{S}_{a,e}\boldsymbol{A}_{xe}^T \qquad (D4)$$

where $\boldsymbol{S}_{a,e}$ is the part of the a priori covariance matrix that corresponds to "extra" state vector elements other than $CO_2$ scale factors. With $N$ total state vector elements and $nlev$ atmospheric levels, $\mathbf{S}_{a,e}$ has dimensions ($N$-$nlev$,$N$-$nlev$). $\boldsymbol{A}_{xe}$ is the subset of the averaging kernel matrix that characterizes the smoothing effect of the extra state vector elements on the $CO_2$ profiles, with dimensions ($nlev$,$N$-$nlev$). The interference error is the smallest contribution to the total error and most of the error comes

from the smoothing effect of the a priori $CO_2$ covariance, followed by the contribution of measurement noise which oscillates between ~10–25% of the a priori $CO_2$ uncertainty. If temperature were retrieved, for example with a temperature offset or with a scale factor added to the extra state vector elements, we would expect the interference error to increase.

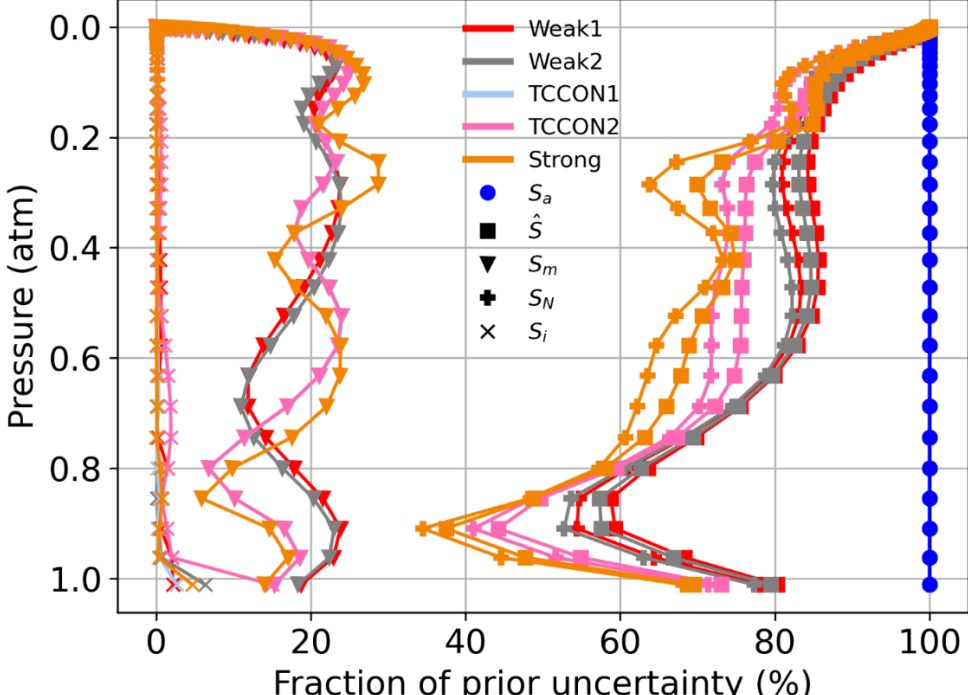

**Figure D8: The square root of the diagonal elements of the retrieval total error covariance matrix $\widehat{S}$, the null space covariance matrix**
**$S_N$, the interference error covariance matrix $S_i$, and the measurement noise error covariance matrix $S_m$ expressed as a fraction of the a priori uncertainty $S_a$.**





## Appendix E: Empirical corrections

In Sect. 3 we saw that $CO_2$ profile retrievals have high sensitivity to $CO_2$ in the absence of errors in the a priori meteorology
and systematic errors in instrument line shape. Here we investigate the possibility of empirically removing the effect of those
errors by de-weighting systematic spectral fitting residuals using empirical orthogonal functions (EOFs). EOFs have been
used, for example, with retrievals from GOSAT and OCO-2 measurements (O'Dell et al., 2018).

### E.1 Empirical orthogonal functions

To reduce the effect of systematic residuals on retrieved profiles, empirical orthogonal functions (EOFs) of the spectral fitting
residuals were derived to find and remove systematic patterns in the residuals related to temperature errors, instrument line
shape, and other effects. The residuals divided by airmass, from a set of retrievals covering a wide range of observational
conditions are stored in a matrix $M(m,n)$ with $n$ the number of spectra and $m$ the number of spectral points. Then a singular
value decomposition is performed on this matrix. The columns of the matrix of left singular vectors are orthogonal basis
vectors of the residuals and those associated with the largest singular values represent the main patterns in the residuals, while
the corresponding right singular vectors can provide information on the temporal frequency of these patterns.

We use a linear combination of left singular vectors. Each singular vector is associated with a scaling factor. The scaling factor
is part of the state vector and adjusted during the retrieval using 100% uncertainty. Before each inversion step, the spectrum
"c" calculated with the forward model becomes:

$$c = c + \sum_{i=1}^{N} a_i u_i \qquad (E1)$$

where $N$ is the number of EOFs to use, ordered with decreasing singular value. The first EOF, associated with the highest
singular value, is like the scaled average residual from all the spectral residuals in the matrix M. Our implementation differs
from that described by O'Dell et al. (2018) in that here the EOFs are derived from a set of residuals obtained using scaling
retrievals, and not using profile retrievals. Since they are meant to remove systematic errors in the calculated spectra before
the retrieval adjusts the $CO_2$ scaling factors, the EOFs should be derived from a large set of residuals obtained with scaling
retrievals to have a significant effect on the profile retrieval. If they are derived from residuals obtained with profile retrievals,
these mainly include systematic error patterns corresponding to interfering species, which are not the main source of deviations
in retrieved $CO_2$ profiles. When using scaling retrieval residuals, each EOF includes different error patterns corresponding to
$CO_2$ absorption lines. These error patterns may be attributed to systematic errors for the first EOF, such as errors in
spectroscopy, or in the instrument line shape, or a persistent bias in meteorology. The error patterns can also correspond to
errors in the a priori meteorology. The temporal frequency of each error pattern is contained in the corresponding right singular
vector. The right singular vectors could help diagnose, for example, biases in a priori temperature profiles on different time





scales. The right singular vectors can also be used to find correlations between each spectral residual patterns and other quantities measured in time, such as differences between a priori and measured meteorology.

If the residual patterns corresponding to $CO_2$ lines have the same shape as residuals caused by errors in the a priori $CO_2$ profile shape, adjustments to the $CO_2$ scaling factors will compete with adjustments to the EOF scaling factors in the retrieval. Because higher order EOFs are associated with residuals with different time periodicity, they can also introduce errors that do not exist in calculated spectra. We chose to only include the first EOF, which represents residual patterns common to most spectra. The leading EOF can explain 40 to 52% of the variability in the residuals, depending on the window, as shown in Fig. E1. The

fraction of variability is obtained as the singular value of a given EOF divided by the sum of all singular values. The first 10 EOFs in each window are above the noise level of singular values and account for over 90% of the variability in the residuals.

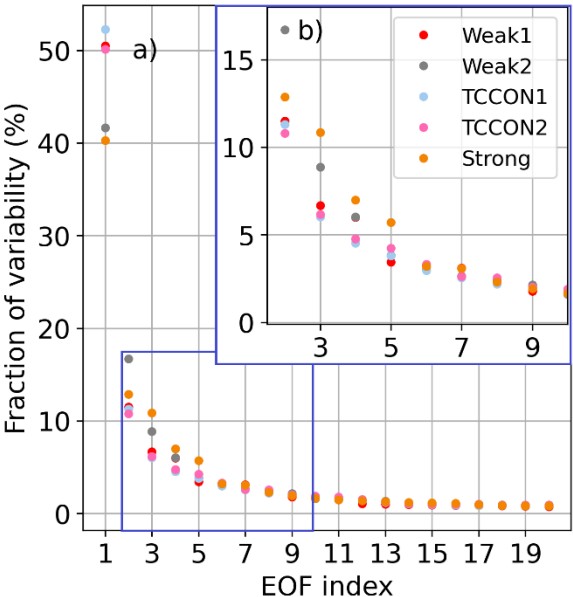

**Figure E1: Fraction of the variability in the spectral residuals accounted for by each empirical orthogonal function in each $CO_2$ window. The EOF numbers are shown in decreasing order of singular value. Panel (b) highlights the blue rectangle inside panel (a).**

**E.2 Results**

One year of measurements from the East Trout Lake (SK, Canada) TCCON station were processed in three ways: with scaling retrievals, with profile retrievals, and with profile retrievals including the first EOF derived from residuals obtained with the scaling retrievals. The residuals used to derive the EOFs are filtered such that spectra that would not pass the TCCON quality checks are not included. To avoid isolated spectra with large residuals to have a disproportionate impact on the singular value

decomposition of the matrix of residuals, all the spectra are ordered by increasing solar zenith angle and filtered based on the root mean square of the residuals: the 500-points rolling median is computed, and the median of the 500-points rolling standard deviation is used as an estimate of the standard deviation σ, then only spectra within 1-σ of the rolling median for all windows





are used to derive the EOFs. The matrix of residuals resulting from this filtering includes 42037 out of 64245 total spectra. $XCO_2$ was retrieved from each window separately. The statistics on the retrieved $XCO_2$ error are shown in Table E1 for each

retrieval type and for each window. In all windows but the Strong window, the changes in $XCO_2$ error between the different retrieval methods are small, less than 0.05 ppm. This is eight times smaller than the reported TCCON 1-σ single-measurement precision of 0.4 ppm. However, the mean $XCO_2$ error is ~55% larger in the strong window with profile retrievals compared to scaling retrievals.

**Table E1: Statistics on the retrieved $XCO_2$ error for one year of measurements at the East Trout Lake TCCON station. "STD" indicates the standard deviation.**

| $XCO_2$ error (ppm) | Scaling retrieval | | | Profile retrieval | | | Profile retrieval with the first EOF | | |
|---|---|---|---|---|---|---|---|---|---|
| Window | Mean | Median | STD | Mean | Median | STD | Mean | Median | STD |
| Strong | 0.51 | 0.38 | 0.37 | 0.79 | 0.63 | 0.60 | 0.78 | 0.61 | 0.59 |
| Weak1 | 0.89 | 0.64 | 0.68 | 0.91 | 0.67 | 0.66 | 0.90 | 0.66 | 0.66 |
| Weak2 | 0.80 | 0.56 | 0.64 | 0.81 | 0.61 | 0.56 | 0.80 | 0.61 | 0.56 |
| TCCON1 | 0.74 | 0.48 | 0.66 | 0.79 | 0.51 | 0.70 | 0.79 | 0.51 | 0.70 |
| TCCON2 | 0.69 | 0.45 | 0.61 | 0.74 | 0.47 | 0.66 | 0.74 | 0.47 | 0.66 |

Figures E2 to E6 show quantities derived from each type of retrieval for an example day and for each window. In each window, the profile retrieval with the first EOF appears as an intermediate case between the profile retrieval and the scaling retrieval.

In each case, the root mean square of the residuals is smaller for profile retrieval with the first EOF, but the $XCO_2$ error is not necessarily smaller.





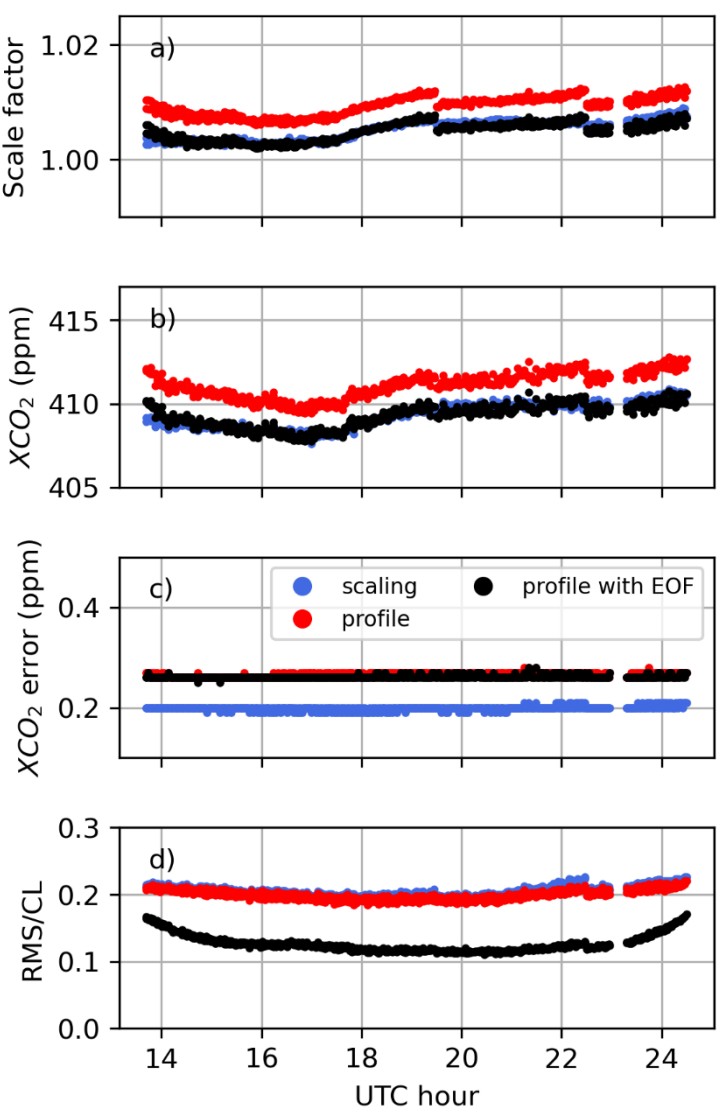

**Figure E2: Quantities derived from retrievals on East Trout Lake measurements on 29 March 2018 for the Strong window. The retrieval type is indicated by the legend. Panel (a) shows the column-integrated $CO_2$ scale factor. Panel (b) shows $XCO_2$ and panel (c) shows the $XCO_2$ error. Panel (d) shows the root mean square of the residuals as a fraction of the continuum level.**





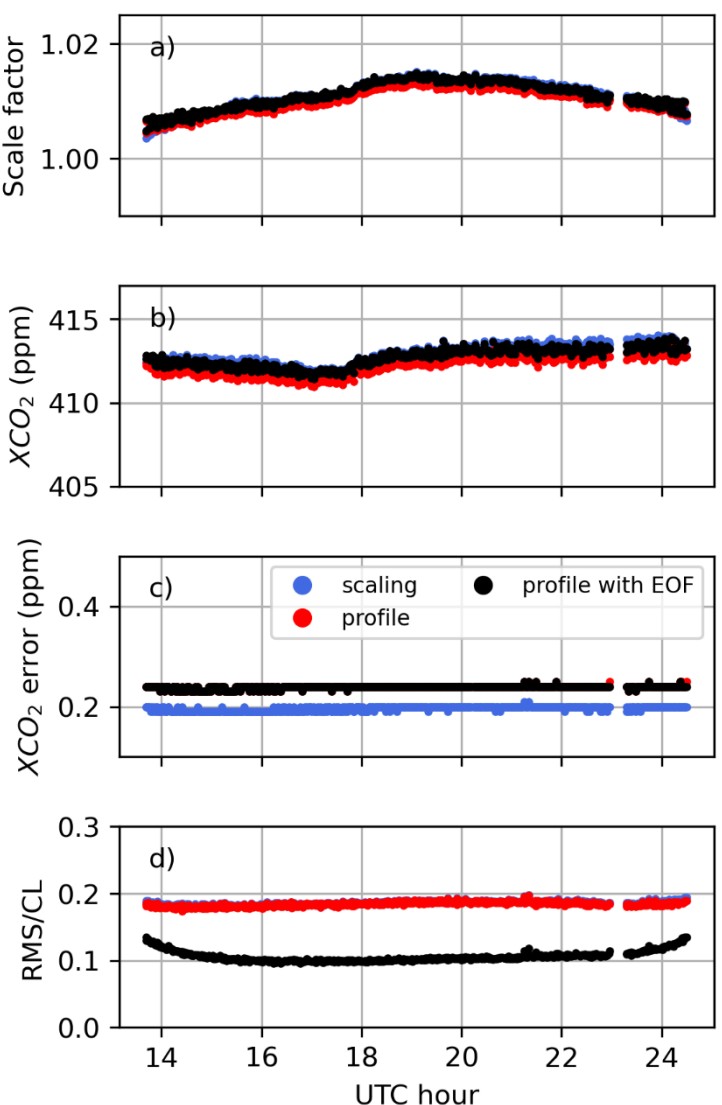

**Figure E3: Same as Fig. E2 but for the TCCON1 window.**





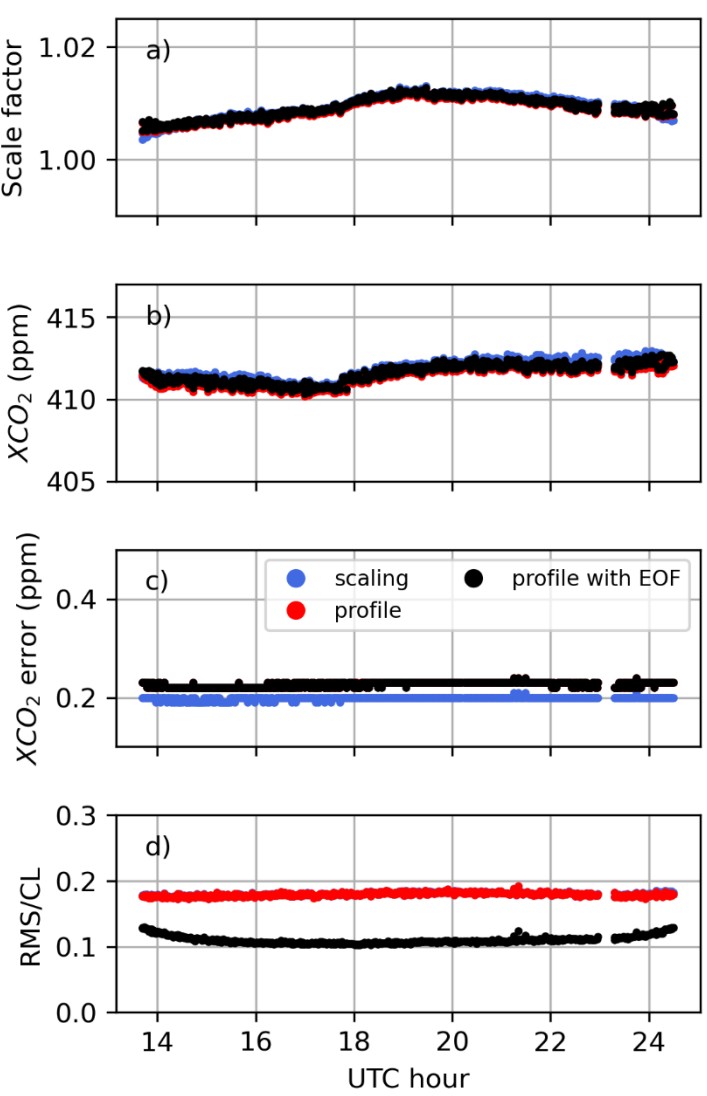


**Figure E4: Same as Fig. E2 but for the TCCON2 window.**





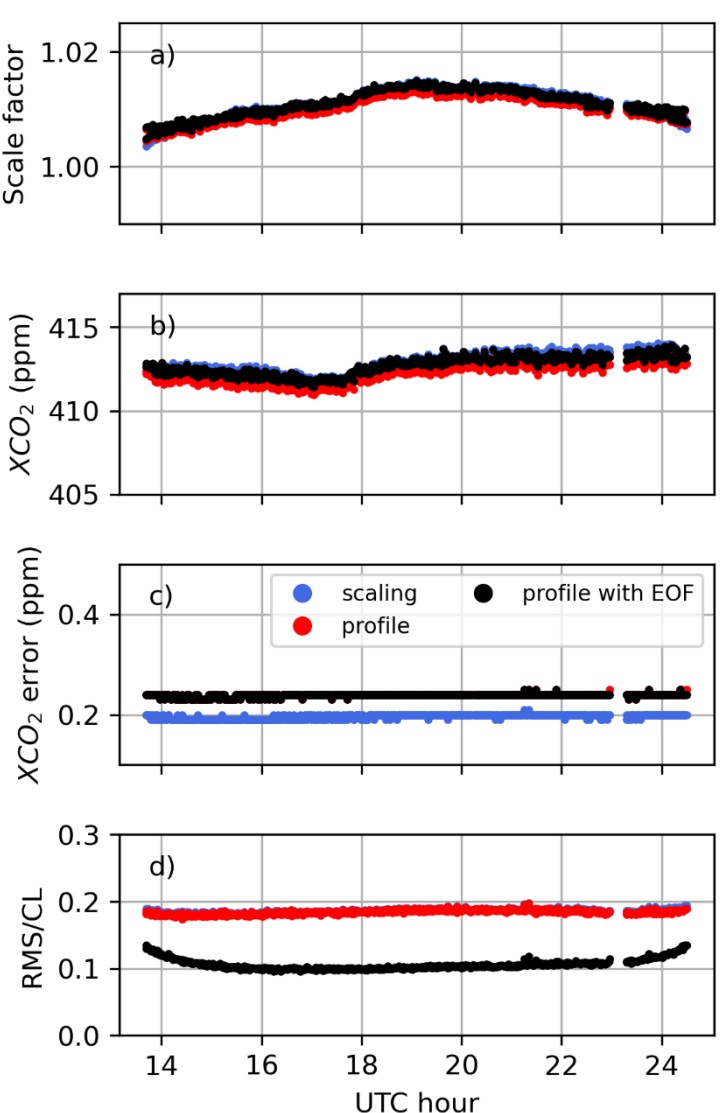

**Figure E5: Same as Fig. E2 but for the Weak1 window.**





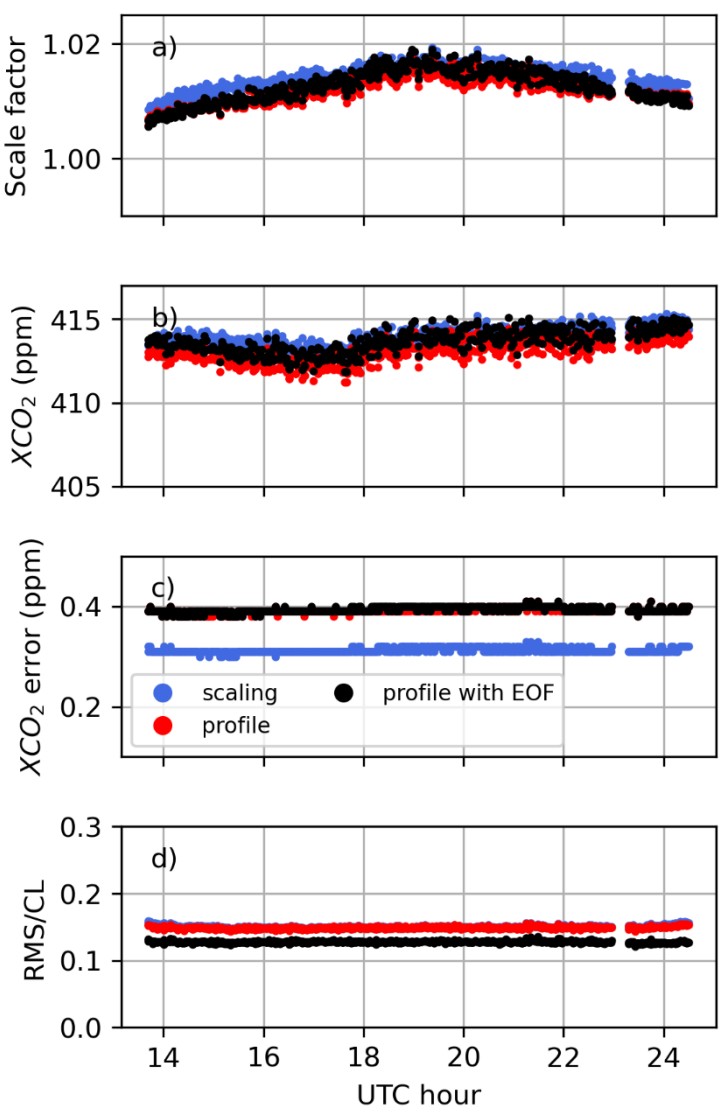

**Figure E6: Same as Fig. E2 but for the Weak2 window.**

In Fig. E7(a), XCO₂ differences are shown between profile and scaling retrievals, and between profile retrievals including the first EOF and scaling retrievals in Fig. E7(b). We have seen that differences in $XCO_2$ error between the different retrieval types are within 0.05 ppm. However, differences in XCO₂ between profile and scaling retrievals can be several times larger than the

$XCO_2\ error$, indicating different sources of bias between profile and scaling retrievals. In the Weak1 window, the median of the XCO₂ absolute differences are ~4 times larger than the median $XCO_2$ error, and ~3 times larger in the Strong window. In the TCCON1, TCCON2, and Weak2 windows, the median of the XCO₂ absolute difference is smaller than the median XCO₂ error. In all but the Weak1 window, the XCO₂ differences are 25 to 35% smaller between August and November than for the



rest of the year. In Fig. E7(b), the $XCO_2$ differences between the profile retrievals with EOF and the scaling retrievals are
smaller and more consistent between windows than in Fig. E7(a). And the median of the $XCO_2$ absolute differences is smaller
than the median $XCO_2$ error in all windows. Including the leading EOF in a profile retrieval reduces the $XCO_2$ differences
between the scaling and profile retrievals, but the $XCO_2$ of the profile retrieval with EOF is more strongly correlated with the
$XCO_2$ of the profile retrieval than that of the scaling retrieval as shown in Table E2.

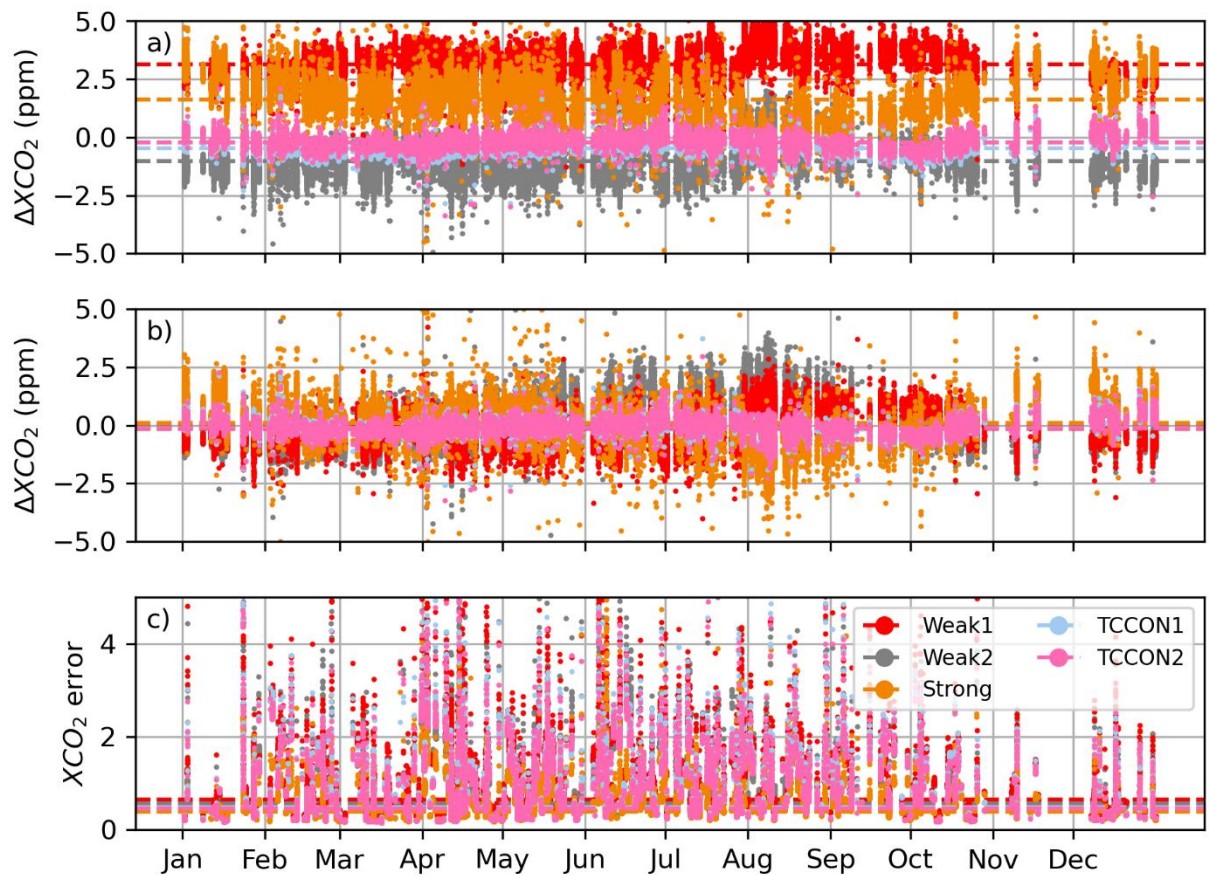

**Figure E7: In panel (a), the $XCO_2$ obtained from the scaling retrieval is subtracted from the $XCO_2$ obtained from the profile
retrieval. In panel (b), the $XCO_2$ obtained from the scaling retrieval is subtracted from the $XCO_2$ obtained from the profile retrieval
with EOF. In panel (c), the $XCO_2$ error from the scaling retrieval is shown, with the median values as dashed lines. In the top two
panels, the horizontal dashed lines show the median values of absolute differences in $XCO_2$.**

When compared to preliminary data from aircraft measurements, the deviations in the $CO_2$ profiles obtained with profile
retrievals are larger than the vertical variations in the aircraft measurement. When the retrieved profiles present large deviations
typical of temperature errors like that in Fig. 8(b), the $CO_2$ profile obtained from profile retrieval with the first EOF reduces
the amplitude of the deviations, but the shape persists. This is expected as the first EOF represents the average residuals, which
should not include residual features caused by temperature errors, unless the temperature errors were always biased in the same





way. We would expect the first EOF to reduce deviations like that in Fig. 8(a). In such cases, the $CO_2$ profiles obtained from

profile retrieval with the first EOF are smoother than profile retrievals but present no clear advantage over scaling retrievals.

**Table E2: Squared Pearson correlation coefficient for $XCO_2$ between the scaling and profile retrievals (SCL–PRF), and between the profile retrieval with the first EOF and the profile retrieval (EOF–PRF).**

| $R^2$ | SCL–PRF | EOF–PRF |
|---|---|---|
| Strong | 0.9368 | 0.9929 |
| Weak1 | 0.9633 | 0.9951 |
| Weak2 | 0.9586 | 0.9814 |
| TCCON1 | 0.9922 | 0.9995 |
| TCCON2 | 0.9931 | 0.9999 |

*Code availability.* It is our intention to make the profile retrieval code available as part of a future version of GGG and hence publicly available through TCCON, until then it can be made available upon request to the corresponding author.

*Data availability.* The data used in this study consists of synthetic spectra generated with GGG2020 and measured spectra from the Lamont and East Trout Lake TCCON stations. Scaling retrieval products from those sites using the GGG2014 algorithm (Wennberg et al., 2016; Wunch et al., 2018) are available publicly through CaltechDATA (https://tccondata.org). Measured solar absorption spectra can be obtained by contacting the TCCON site PIs.

*Competing interests.* The authors declare that they have no conflict of interest.

*Author contributions.* SR analyzed the data and prepared the manuscript with detailed feedback from all co-authors. JM implemented qSDV+LM into GFIT and provided guidance for its use with GFIT2. BJC wrote the initial GFIT2 algorithm and SR implemented changes relevant to this study. GCT is the author of GGG and provided support and significant conceptual input with DW, KS, and BJC. JLL and SR developed the code to generate the GGG2020 a priori profiles. The East Trout Lake solar spectra were provided by DW and JM. CS and BB provided the AirCore data. SCB provided the Lamont surface and aircraft measurements.

*Acknowledgements.* This work was funded by the Natural Sciences and Engineering Research Council of Canada (grant 433842-2012), and the Canadian Space Agency (contract 45-7014551). The GEOS-5-FPIT data used in this study have been provided by the Global Modeling Assimilation Office (GMAO) at NASA Goddard Space Flight Center. The Lamont solar spectra were provided by Coleen Roehl and Paul Wennberg from the California Institute of Technology. The Lamont TCCON site operates with funding from NASA. In situ observations and vertical profiles collected over the Southern Great Plains were





supported by the Office of Biological and Environmental Research of the US Department of Energy under contract no. DE-AC02-05CH11231 as part of the Atmospheric Radiation Measurement (ARM) Program, ARM Aerial Facility (AAF), and
Terrestrial Ecosystem Science (TES) Program. The GNU parallel package (Tange, 2011) has been used to run tests with GFIT2.

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
