# Peer review of "Retrieval of atmospheric CO2 vertical profiles from ground-based near-infrared spectra"

_Atmospheric Measurement Techniques, 2020_

## Editor Comment (EC1) · Frank Hase (Editor) · 13 Nov 2020

**Comment on "Retrieval of atmospheric $CO_2$ vertical profiles from ground-based near-infrared spectra".**

By Frank Hase

The authors address a very relevant topic in their investigation, as it would be highly desirable to gain the capability of deriving vertical mixing ratios of $CO_2$ from high-resolution ground-based solar absorption spectra as recorded by TCCON. It needs to be noted, that the involved researchers at University of Toronto work enduringly since many years on improvements of the required spectroscopic data sets, which is an important pre-requisite for developing successful profile retrievals of atmospheric carbon dioxide.

However, I would like to add a few technical comments on the manuscript. Perhaps the authors might find these suggestions useful for further enhancing this study.

Page 2, line 66: "Scaling retrievals do not require inter-level constraints on a–priori concentration uncertainties" – I agree that a scaling retrieval does not require the explicit construction of inter-level constraints, but actually scaling equals the assumption of very strong inter-level constraints.

I would suggest to rephrase the notion on how GFIT handles a scaling retrieval (I hope I understand the authors correctly), as, e.g. "Technically, GFIT handles the scaling retrieval by weakly constraining the fitted VSF factor. The approach is equivalent to performing an optimal estimation of the VSF, assigning a value of unity to the a-priori VSF and a value of 1e6 as its expected range of variability."

Page 3, line 96: the line mixing referred to by the authors is (I believe) Rosenkranz line mixing and should be referred to as such.

Page 6, line 147: "We see no advantage to fitting non-contiguous windows in parallel, rather than in series, and then averaging the results." In my opinion, this is a misjudgement. Especially for retrieving profile information, combining weak and strong bands in a simultaneous fit is known to be potentially very advantageous. While the line wings of saturated lines in a strong band carry information about the lowermost atmospheric layers, weaker lines contribute information on higher atmospheric levels. We tried combination of bands in the context of the cited work by Dohe, which improved the uniformity of partial column sensitivities significantly over what is shown in Fig. 8.3 in the work of Dohe (using only the strong band). At that time, spectroscopic inconsistencies hindered a successful combined fit of weak and strong bands, given the progress on spectroscopic data this might look different today. In any case, the general statement that such a capability does not offer an advantage is in my opinion highly questionable. (The method of sequential estimation would be equivalent to a combined fit (if linearity can be assumed) and could be used as a makeshift solution if a fit of several windows is not supported by the code, but I assume this is not what the authors describe by "averaging the results".)

Section 3.1: In my feeling, this section would better correspond to the following investigation using measured spectra if a similar (not much higher) SNR and a similar and more realistic a-priori covariance would be used. I miss a sensitivity study concerning ILS in section 3.1 (this should be possible to realize although the code does not yet explicitly support ILS parameters by using slightly different acceptance cones in the forward calculation and in the retrieval, thereby modelling a modulation loss via the self-apodisation). I also miss a sensitivity study with respect to the offset (as it cannot be fitted in the non-saturated bands). By the way: as a DC-correction is implemented and as the non-linearity is well controlled (I assume), why is an offset observed in the saturated band? Does this offset behave like noise – varying from spectrum to spectrum – or is this a systematic offset? It would be interesting to provide a figure showing the behaviour of the fitted offset for a larger number of spectra (could be included in fig E2?).

I find the fact that the fit residuals of the unperturbed spectrum shows these oscillatory high-resolution features in the residuals very irritating. Why is it not possible to achieve a self-consistent code configuration that applies the same approximations in the simulation of the spectrum as in the forward calculation performed as part of the retrieval? Note that for the disturbance of line parameters (panels e and f in figure 7), these artificial features seem to be of similar size as those invoked by the perturbation.

Figure 11: it might be instructive to add a further panel displaying $XCO_2$ for each retrieval setup.

Figure 13: Why is GGG XCO2 so significantly off from the in situ value (scaling retrievals in the TCCON bands are off by about 4 ppm)? I assume that required calibration factors have not yet been applied?

Note that this problem of bias is connected to a significant problem in your retrieval setup: as far as I oversee, the $CO_2$ a-priori profiles used are intended to match with the actual $CO_2$ mixing ratios in the atmosphere. But the $CO_2$ becomes calibrated to WMO scale only further down the processing line (by rationing $CO_2$ column over the $O_2$ column and application of ADCFs + AICFs). Let's assume the calibration is off: this poses no problem for a scaling retrieval (same residuals, same solution, only a different scaling factor is reported). However, an optimal estimation setup will misbehave if the a-priori is biased: it will try to approach the a-priori values in altitudes where the sensitivity is low and in effect the retrieved profile will oscillate significantly. I think it would be superior to apply a pure smoothness constraint as used by Dohe instead of an optimal estimation with diagonal constraint and just some correlation on top. If a pure smoothness constraint is used, a bias of the a-priori remains without effect  even for a profile retrieval (same residuals, same solution, only different scaling factors on identical profile shapes). Moreover, this setup is preferable because it allows a smooth transition between the profile retrieval and the simple scaling retrieval (just increase the smoothing constraint further and further).

---

## Author Comment (AC1) · 22 Dec 2020

Response to the Editor's Comment (doi:10.5194/amt-2020-429-EC1) on "Retrieval of atmospheric CO2 vertical profiles from ground-based near-infrared spectra" by Sébastien Roche et al.

We thank Frank Hase for his thoughtful comments. Below we address each comment and the resulting revisions to the text and figures will be added to the final revised manuscript which will also include revisions based on comments from the referees.

**Comment 1: scaling retrieval and interlayer constraints**

Page 2, line 66: "Scaling retrievals do not require inter-level constraints on a-priori concentration uncertainties" – I agree that a scaling retrieval does not require the explicit construction of inter-level constraints, but actually scaling equals the assumption of very strong inter-level constraints. I would suggest to rephrase the notion on how GFIT handles a scaling retrieval (I hope I understand the authors correctly), as, e.g. "Technically, GFIT handles the scaling retrieval by weakly constraining the fitted VSF factor. The approach is equivalent to performing an optimal estimation of the VSF, assigning a value of unity to the a-priori VSF and a value of 1e6 as its expected range of variability."

**Response to comment 1**

It was indeed meant that the scaling retrieval does not need explicit interlayer correlations.

The text will be updated as suggested.

**Comment 2: line mixing**

Page 3, line 96: the line mixing referred to by the authors is (I believe) Rosenkranz line mixing and should be referred to as such.

**Response to comment 2**

Yes, this is the Rosenkranz approximation for line mixing, the text will be updated to:

This version of the code implements quadratic speed-dependent Voigt line shapes with line mixing (qSDV+LM) for CO2 (Mendonca et al., 2016) and CH4 (Mendonca et al., 2017) bands, and qSDV line shapes for O2 in the band centered at 1.27  $\mu$ m (Mendonca et al., 2019). The line mixing coefficients are derived with the first order Rosenkranz approximation (Rosenkranz et al., 1975).

With the added reference:

Rosenkranz, P.: Shape of the 5 mm oxygen band in the atmosphere, IEEE Trans. Antennas Propag., 23(4), 498–506, doi:10.1109/TAP.1975.1141119, 1975.

**Comment 3: sequential vs parallel retrievals**

Page 6, line 147: "We see no advantage to fitting non-contiguous windows in parallel, rather than in series, and then averaging the results." In my opinion, this is a misjudgement. Especially for retrieving profile information, combining weak and strong bands in a simultaneous fit is known to be potentially very advantageous. While the line wings of saturated lines in a strong band carry information about the

lowermost atmospheric layers, weaker lines contribute information on higher atmospheric levels. We tried combination of bands in the context of the cited work by Dohe, which improved the uniformity of partial column sensitivities significantly over what is shown in Fig. 8.3 in the work of Dohe (using only the strong band). At that time, spectroscopic inconsistencies hindered a successful combined fit of weak and strong bands, given the progress on spectroscopic data this might look different today. In any case, the general statement that such a capability does not offer an advantage is in my opinion highly questionable. (The method of sequential estimation would be equivalent to a combined fit (if linearity can be assumed) and could be used as a makeshift solution if a fit of several windows is not supported by the code, but I assume this is not what the authors describe by "averaging the results".)

**Response to comment 3**

It is what we were describing, we should not have used the word "average". We were referring to the method of retrieving a profile from each window separately, and combining the resulting profiles in post processing by taking into account the jacobian of each window. We have not shown such combined profiles, they tend to be heavily weighted towards the profile retrieved from the Strong window. The combined averaging kernel indicates better sensitivity than any single window, but the combined profile is as affected by biases caused by temperature errors as the Strong window profiles. This should also be the case for parallel retrievals.

A simultaneous retrieval could be an advantage for gases for which the problem is more non-linear, where the change in the jacobian computed with the a priori state compared to the jacobian computed with the retrieved state is relatively large. But CO2 is a special case as the a priori profiles compare well with the true profiles.

A sequential retrieval also gives the ability to diagnose potential issues between the different windows.

**Comment 4: sensitivity study**

Section 3.1: In my feeling, this section would better correspond to the following investigation using measured spectra if a similar (not much higher) SNR and a similar and more realistic a-priori covariance would be used. I miss a sensitivity study concerning ILS in section 3.1 (this should be possible to realize although the code does not yet explicitly support ILS parameters by using slightly different acceptance cones in the forward calculation and in the retrieval, thereby modelling a modulation loss via the selfapodisation). I also miss a sensitivity study with respect to the offset (as it cannot be fitted in the nonsaturated bands). By the way: as a DC-correction is implemented and as the non-linearity is well controlled (I assume), why is an offset observed in the saturated band? Does this offset behave like noise – varying from spectrum to spectrum – or is this a systematic offset? It would be interesting to provide a figure showing the behaviour of the fitted offset for a larger number of spectra (could be included in fig E2?). I find the fact that the fit residuals of the unperturbed spectrum shows these oscillatory high-resolution features in the residuals very irritating. Why is it not possible to achieve a self-consistent code configuration that applies the same approximations in the simulation of the spectrum as in the forward calculation performed as part of the retrieval? Note that for the disturbance of line parameters (panels e and f in figure 7), these artificial features seem to be of similar size as those invoked by the perturbation.

**Response to comment 4**

**Constraints**

The high SNR and loose prior constraint used in Sect. 3.1 highlights the source of variability in the retrieved profiles by forcing the retrieval to pull most of the information from the measurement. The section shows that including smoothness constraints on the  $CO_2$  profile will mainly serve to reduce variability that is mainly caused by errors in the a priori temperature profile. Here we want to be able to identify the sources of variability to indicate which parts of the forward model should be improved in priority. The magnitude of scaling retrieval residuals caused by typical temperature errors is much larger that caused by incorrect  $CO_2$  profile shape. Without a temperature retrieval scheme, the adjustment to the  $CO_2$  profile in a profile retrieval will be mainly driven by variability in the temperature profile.

This is a critical point in the paper. The prior constraint should not be adjusted unless oscillations in the retrieved profiles caused by typical errors in sources other than  $CO_2$  are smaller than typical  $CO_2$  variability even when using a diagonal a priori covariance matrix.

**ILS and Zero-level offset**

Below a type of ILS error is considered (widening by perturbing the internal field of view diameter), and the effect of a zero-level offset is shown.

Figure R1(c) and R1(d) show the effect of perturbing the internal FOV diameter by +7%, which leads to a widening of the ILS. The deviations from the truth are within 1 ppm for P > 0.5 atm and within 3 ppm for P

---

## Referee Comment (RC1) · Anonymous Referee #1 · 4 Feb 2021

Summary

The TCCON network is an immensely valuable WMO-calibrated record of CO2 essential for validation of satellite CO2 columns (as well as other trace gases) and has additional use in the study of the carbon cycle.

Currently TCCON is used for validation of satellite full-column products in the near-infrared, e.g. XCO2 products from OCO-2/3, GOSAT/2, Tansat, and future planned missions will also rely on the TCCON network. It would be immensely valuable if TCCON had a vertically resolved product that could be used to validate vertically-resolved satellite products, e.g. thermal infrared CO2 from AIRS or GOSAT/2, the GOSAT NIR/TIR profile, the lower tropospheric research products from GOSAT/2 or OCO-2/3, and to study the carbon cycle.

[Figure]

Overall comments

The paper was well laid out and provides useful and important information about the sources of errors in different TCCON and satellite NIR bands. It would be useful to see results of joint-band retrievals, however accept this limitation of GFIT. The two things that are missing are:

1) predicted errors propagated from the interferent errors, These errors are calculated empirically and should also be compared to their predicted impact, at least for temperature and water. These errors can be calculated using Jacobians, following Connor et al. (2016) Eq. 6, or Worden et al. (2004) (second term of Eq. 18).

2) The performance of the profile retrieval with real spectra is only reported as a column result (e.g. Fig 13). The standard deviation and bias versus AirCore should be calculated at all pressures and summarized in the paper.

Specific comments

Line 17. To help the reader understand the scope of the paper, I would change the second sentence of the abstract to, "With these improvements, CO2 profiles were obtained from sequential retrievals in five spectral windows with different vertical sensitivities using synthetic and real spectra."

Add to the paper: It was unclear to me (or maybe I missed it) what is retrieved. Is surface pressure? Water profile? Water scaling? Temperature profile? Temperature scaling? This could be listed at the start of methods.

Line 82. I would add, "... and would also allow TCCON to be used for validation of TIR satellite products, e.g. from AIRS and GOSAT/2, and vertically resolved NIR GOSAT and OCO-2 experimental products."

Line 117. I would add a sentence saying, "OCO-2/3 and GOSAT/2 use the Weak1 and Strong bands as well as a band centered at 0.765 um not used by TCCON."

Line 147. The sentence, "We see no advantage to fitting noncontiguous windows in parallel, rather than in series, and then averaging the results." is incorrect. Mathematically these are very different. The way to fit sequentially is called a sequential update and discussed in Rodgers, 2000 or Dudhia et al., 2002 and involves setting the constraint for the next step to the error covariance from the previous step.

Line 189. It would help the reader to have the value for sigma listed for 100, 500, 200, and 1 hPa.

Line 194. A 2 km length scale seems very narrow, particularly above 4 km. Just a comment.

Line 228. Add a sentence, "The DOFS are shown in figures 3-6, and 8-9."

Figures 3-5. These errors should be compared to the predicted errors.

Figures 3-5. Question that might be out of the scope of this paper: do these errors affect TCCON XCO2? In other words can the same tests from Figures 3-5 be done for the standard TCCON retrieval?

Figure 8. I do not understand how the result can be up to 50 ppm off when the a priori is set to 6 ppm.

Table 5. My note: I have not found information content to be useful because it is not aware of many of the systematic errors in the system.

Figure 13. My note: The variability of the results for xco2 for the different bands was surprising.

References

Dudhia, A., V. L. Jay, and C. D. Rodgers (2002), Microwindow selection for high-spectral-resolution sounders, Appl. Opt., 41(18), 3665–3673.

Rodgers, C. D. (2000), Inverse Methods for Atmospheric Sounding: Theory and Prac-

tice, World Sci., River Edge, N. J.

Worden, J., Kulawik, S., Shepard, M., Clough, S., Worden, H., Bowman, K., and Goldman, A.: Predicted errors of tropospheric emission spectrometer nadir retrievals from spectral window selection, J. Geophys. Res.-Atmos., 109, D09308, https://doi.org/10.1029/2004JD004522, 2004.

Connor, B., Bösch, H., McDuffie, J., Taylor, T., Fu, D., Frankenberg, C., O'Dell, C., Payne, V. H., Gunson, M., Pollock, R., Hobbs, J., Oyafuso, F., and Jiang, Y.: Quantification of uncertainties in OCO-2 measurements of XCO2: simulations and linear error analysis, Atmos. Meas. Tech., 9, 5227–5238, https://doi.org/10.5194/amt-9-5227-2016, 2016.

---

## Referee Comment (RC2) · Anonymous Referee #2 · 8 Feb 2021

Generally,

The authors evaluate the CO2 vertical profile retrieval from the NIR spectra recorded at TCCON sites using the GFITT2 algorithm. The retrieval uncertainty is discussed in terms of temperature, a priori profiles, instrument alignment, and spectroscopic parameters. Although the reliable CO2 profile retrieval remains difficult, this study does the pilot study and shows some interesting results. Overall, I recommend the publication of this manuscript after addressing the following comments.

Specific comments

Line 26-28: please add 'at Lamont' at the end of the sentence.

Line 61: "non-linearleast-squares spectral fitting algorithm " is GFIT also based on the

[Figure]

OEM? If yes, please mention it in the text. If not, please add some discussion about the retrieval method in the paper.

Line 62: "A forward model . . . a priori knowledge of atmospheric conditions", please also add instrumental conditions and the observation geometry.

Line 66: "Scaling retrievals do not require inter-level constraints on a priori concentration uncertainties. ". A scaling retrieval is actually equal to a very strong inter-level constraint. Please re-write this sentence.

Line 67. "an uncertainty of 10ˆ6 ". Please check why the uncertainty is so large?

Line 101. Does the wavenumber scales not included in GGG2014? If not, please mention it in the text.

Line 107: change "original TCCON spectral windows " to "original TCCON retrieval windows"

Line 148:" We see no advantage to fitting non- contiguous windows in parallel, rather than in series, and then averaging the results. ". How do you average the results? Do you apply the weighting function based on the SNR of each window?

Line 312-314:As the GFIT2 does not allow retrieve 2 profiles simultaneously, have you ever tried to retrieve the H2O profile first, and use the retrieved H2O profile as the a priori profile to do the CO2 retrieval?

Line 444: What is the physical reason for the 'zero-level offset' error?

Line 445: What do you mean by "higher altitudes "? please note the vertical range specifically.

Line 545. Section 3.2.4, I expect the authors to present a table here summarizing all the uncertainties for the CO2 profile at Lamont, together with the typical vertical variation of CO2.

---

## Author Response (AR1)

**Reponse to reviews**

We thank the Editor and Referees for their thoughtful comments and questions.

In this document we present our responses to the Editor Comment (EC1) and to the comments from the two anonymous referees (RC1 and RC2). Each response is followed by the corresponding change in the text, with the updated text highlighted in red and any removed text shown with a strikethrough.

**Spontaneous updates**

Some of the number of coincident spectra shown in Table 3 were off by 1.

**The caption of Table 5 was updated to:**

Shannon information content (H), degrees of freedom for signal (DOFS) for the CO2 profile, and Ratio of Residuals (RR) averaged over all 492 profile retrievals from near-infrared TCCON spectra measured at Lamont and coincident within ±1 h of the AirCore last sampling time. The standard deviation is also shown.

**Correction in appendix C:**

$$\boldsymbol{c}_s = \mathbf{A}_{col}(\boldsymbol{c} - \boldsymbol{c}_a) + \boldsymbol{c}_a \qquad (C6)$$

where $\boldsymbol{c}_s$ is the smoothed partial column profile, $\boldsymbol{c}$ is the  partial column profile to be smoothed, and $\boldsymbol{c}_a$ is the a priori partial column profile. Or using the total column averaging kernel:

**Correction in appendix D:** Changed $S_e$ to $S_y$ in equations D2 and D3 to be consistent with the notations in the rest of the document.

**Response to EC1**

**Comment #1:**

Page 2, line 66: "Scaling retrievals do not require inter-level constraints on a–priori concentration uncertainties" – I agree that a scaling retrieval does not require the explicit construction of inter-level constraints, but actually scaling equals the assumption of very strong inter-level constraints. I would suggest to rephrase the notion on how GFIT handles a scaling retrieval (I hope I understand the authors correctly), as, e.g. "Technically, GFIT handles the scaling retrieval by weakly constraining the fitted VSF factor. The approach is equivalent to performing an optimal estimation of the VSF, assigning a value of unity to the a-priori VSF and a value of 1e6 as its expected range of variability."

**Response:**
It was indeed meant that the scaling retrieval does not need explicit interlayer correlations.
The text will be updated as suggested.

**Update:**

Technically, GFIT handles the scaling retrieval by weakly constraining the fitted VSF factor. The approach is equivalent to performing an optimal estimation of the VSF, assigning a value of unity to the a-priori

VSF and a value of 106 as its expected range of variability.

**Comment #2:**
Page 3, line 96: the line mixing referred to by the authors is (I believe) Rosenkranz line mixing and should be referred to as such.

**Response:**
Yes, this is the Rosenkranz approximation for line mixing

**Update:**
This version of the code implements quadratic speed-dependent Voigt line shapes with line mixing (qSDV+LM) for $CO_2$ (Mendonca et al., 2016) and $CH_4$ (Mendonca et al., 2017) bands, and qSDV line shapes for $O_2$ in the band centered at 1.27 µm (Mendonca et al., 2019). The line mixing coefficients are derived with the first order Rosenkranz approximation (Rosenkranz et al., 1975).

Added reference:
Rosenkranz, P.: Shape of the 5 mm oxygen band in the atmosphere, IEEE Trans. Antennas Propag., 23(4), 498–506, doi:10.1109/TAP.1975.1141119, 1975.

**Comment #3:**
Page 6, line 147: "We see no advantage to fitting non-contiguous windows in parallel, rather than in series, and then averaging the results." In my opinion, this is a misjudgement. Especially for retrieving profile information, combining weak and strong bands in a simultaneous fit is known to be potentially very advantageous. While the line wings of saturated lines in a strong band carry information about the lowermost atmospheric layers, weaker lines contribute information on higher atmospheric levels. We tried combination of bands in the context of the cited work by Dohe, which improved the uniformity of partial column sensitivities significantly over what is shown in Fig. 8.3 in the work of Dohe (using only the strong band). At that time, spectroscopic inconsistencies hindered a successful combined fit of weak and strong bands, given the progress on spectroscopic data this might look different today. In any case, the general statement that such a capability does not offer an advantage is in my opinion highly questionable. (The method of sequential estimation would be equivalent to a combined fit (if linearity can be assumed) and could be used as a makeshift solution if a fit of several windows is not supported by the code, but I assume this is not what the authors describe by "averaging the results".)

**Response:**
It is what we were describing, we should not have used the word "average". We were referring to the method of retrieving a profile from each window separately, and combining the resulting profiles in post processing by taking into account the jacobian of each window. We have not shown such combined profiles, they tend to be heavily weighted towards the profile retrieved from the Strong window. The combined averaging kernel indicates better sensitivity than any single window, but the combined profile is as affected by biases caused by temperature errors as the Strong window profiles. This should also be the case for parallel retrievals.
A simultaneous retrieval could be an advantage for gases for which the problem is more non-linear, where the change in the jacobian computed with the a priori state compared to the jacobian computed with the retrieved state is relatively large. But $CO_2$ is a special case as the a priori profiles compare well with the true profiles.

A sequential retrieval also gives the ability to diagnose potential issues between the different windows.

**Update:**
We see no advantage to fitting non-contiguous windows simultaneously in parallel, rather than separately in series, and then combining averaging the results. In TCCON post-processing the total columns retrieved from different retrieval windows ($CO_2$ from the TCCON1 and TCCON2 windows, for example) are averaged after removing window-dependent multiplicative biases, using retrieval errors as weights.

**Comment #4:**

Section 3.1: In my feeling, this section would better correspond to the following investigation using measured spectra if a similar (not much higher) SNR and a similar and more realistic a-priori covariance would be used. I miss a sensitivity study concerning ILS in section 3.1 (this should be possible to realize although the code does not yet explicitly support ILS parameters by using slightly different acceptance cones in the forward calculation and in the retrieval, thereby modelling a modulation loss via the selfapodisation). I also miss a sensitivity study with respect to the offset (as it cannot be fitted in the nonsaturated bands). By the way: as a DC-correction is implemented and as the non-linearity is well controlled (I assume), why is an offset observed in the saturated band? Does this offset behave like noise – varying from spectrum to spectrum – or is this a systematic offset? It would be interesting to provide a figure showing the behaviour of the fitted offset for a larger number of spectra (could be included in fig E2?). I find the fact that the fit residuals of the unperturbed spectrum shows these oscillatory high-resolution features in the residuals very irritating. Why is it not possible to achieve a self-consistent code configuration that applies the same approximations in the simulation of the spectrum as in the forward calculation performed as part of the retrieval? Note that for the disturbance of line parameters (panels e and f in figure 7), these artificial features seem to be of similar size as those invoked by the perturbation.

**Response:**
The high SNR and loose prior constraint used in Sect. 3.1 highlights the source of variability in the retrieved profiles by forcing the retrieval to pull most of the information from the measurement. The section shows that including smoothness constraints on the CO2 profile will mainly serve to reduce variability that is mainly caused by errors in the a priori temperature profile. Here we want to be able to identify the sources of variability to indicate which parts of the forward model should be improved in priority. The magnitude of scaling retrieval residuals caused by typical temperature errors is much larger that caused by incorrect CO2 prior profile shape. Without a temperature retrieval scheme, the adjustment to the CO2 profile in a profile retrieval will be mainly driven by variability in the temperature profile.
This is a critical point in the paper. The prior constraint should not be adjusted unless oscillations in the retrieved profiles caused by typical errors in sources other than CO2 are smaller than typical CO2 variability even when using a diagonal a priori covariance matrix.

A sensitivity test for a perturbed FOV and ZLO was be added as Appendix F. It shows the effect of typical zero-level offsets will not be a major source of variability in the retrieved profiles in the Weak and TCCON windows, at least not in the Lamont data used here. If the zero-level offset obtained from the Strong window with real spectra is added in the TCCON and Weak windows before the retrieval, the change in the retrieved profiles is less than 3 ppm at all altitudes for the Lamont spectra.

In GFIT, the measured spectrum is convolved with the truncated and windowed instrument function, but with the rectangular part of the ILS having 0 width. This improves the agreement between measured and calculated spectra. But this process was also applied when using synthetic spectra as observations and lead to the high frequency residuals observed in the reference case in Figure 7 of the preprint, which correspond to differences in the synthetic spectrum before and after the convolution. Not applying this process for synthetic spectra removes most of the residuals in the reference case.
Figure R1 below shows the effect on profiles retrieved for the reference case when no perturbation is applied. Profiles become closer to the priori, within less than 0.3 ppm for the Weak and Strong windows, and within less than 0.01 ppm in the TCCON windows. The fits to synthetic spectra are still not perfect, but the residuals in the reference case are now an order of magnitude smaller than residuals obtained when perturbing the spectroscopic parameters.
All figures of Section **3.1 Synthetic Spectra** were updated following this fix. It only has a visible effect on retrieved profiles in the reference case and does not affect the results and conclusion of this section.

[Figure]

**Figure R1: profiles retrieved from fits to synthetic spectra with no perturbations. In (a) with the extra convolution of the synthetic spectrum with the ILS, and in (c) without**

**Update:**
Paragraph added before section 3.1.1 in **3.1 Syntehtic spectra**:
The total retrieval random error for the retrievals presented in this section is ~4.5% (~18 ppm), the contribution of random noise is ~0.8% (~3 ppm), see Appendix D for definitions of total and measurement noise errors. When the deviations from the truth are larger than the a priori uncertainty

(~20 ppm), it means the perturbation applied has a severe effect on the retrieval. Of course this can be mitigated by using a stronger a priori constraint or a measurement covariance matrix that reflects expected systematic errors, and not just random noise, but always at the cost of reduced sensitivity to $CO_2$ too. The goal here is to estimate the relative effect of different kinds of expected systematic errors on retrieved profile shapes. Stronger constraints can only reduce the amplitude of the deviations from the truth, but the same structures would remain. When the perturbation to a parameter other than $CO_2$ results in deviations from the truth much larger than those presented in Section 3.1.1, it means that errors in that parameter will dominate the variability in the retrieved $CO_2$ profiles regardless of the retrieval constraints.

Added appendix F:
**Appendix F: Synthetic spectra, perturbed field of view and zero-level offset.**

The saturated lines of the Strong window allow to fit a zero-level offset. Figure F1 shows the zero-level offset retrieved from the Strong window using real spectra for each of the days with Lamont data used in Sect. 3. The median absolute value is at most 0.001 on 23 July 2013. The effect of a zero-level offset on retrieved profiles was tested with synthetic spectra. Figure F2(a) and F2(b) are the same as Fig. 4(a) and (b) and show profiles retrieved from synthetic spectra in the reference case, when no perturbation is applied. Figure F2(e) and F2(f) show the effect of a +0.002 perturbation to the zero-level offset, without retrieving it in the Strong window. This has a large effect in the profile retrieved from the Strong window, with deviations from the truth within 30 ppm, and a smaller effect in the other bands with deviations up to 10 ppm.

In Fig. F2(c) and F2(d) we also consider the effect of one type of ILS error by perturbing the internal field of view by +7%, this leads to a widening of the ILS. The unperturbed internal field of view of the spectrometer is 2.4 mrad. The deviations from the truth are within 1 ppm for P > 0.5 atm and within 3 ppm for P < 0.5 atm.

This sensitivity test shows the effect of zero-level offsets will not be a major source of variability in the retrieved profiles. If the zero-level offset retrieved from the Strong window is added to the TCCON and Weak windows before the retrieval, the change in the retrieved profiles is less than 3 ppm at all altitudes as shown in Fig. F3 using days with AirCore profiles at Lamont.

[Figure]

**Figure F1: Zero-level offset retrieved from the Strong CO₂ window for the Lamont spectra coincident within ±1 hour of the last AirCore sampling time and within ±1.5 hour of the closest a priori time on each of the days indicated by the legend. The dashed lines mark the median value for each date.**

[Figure]

**Figure F2:** The left-hand panels show $CO_2$ profiles retrieved using synthetic spectra. In (a), we use the AirCore profile, which was used to generate the synthetic spectra, as the a priori. In (c), the internal field of view is perturbed by +7%, increasing the width of the ILS. In (e), the zero-level offset is perturbed by +0.002 and is not retrieved in the Strong window. The right-hand panels: (b), (d), and (f), show the difference between the retrieved profiles and AirCore, corresponding to (a), (c), and (e) respectively.

[Figure]

**Figure F3: Using real Lamont spectra with the AirCore profile as a priori, the zero-level offset was first retrieved from the Strong window and then added in the Weak and TCCON windows. The difference in the retrieved profiles with and without the added offset is shown for each window and for all the days with AirCore profiles over Lamont. In the Strong window, where the offset is retrieved, the differences are less than 0.001 ppm.**

Added to **3.1.5 Syntehtic spectra: discussion**:
The effect of an error in the instrument's internal field of view and the effect of a zero-level offset are presented in Appendix F, both should lead to minor deviations from the truth, within less than 3 ppm.

Added in **3.2.1 Profiles**:

~~The cause of the deviations in Fig. 8(a) and 9(a) could be due to errors in the zero-level offset in the Weak and TCCON windows, where it is assumed to 445 be zero. In these windows the zero-level offset is not fitted as they lack saturated absorption lines. Errors in the instrument line shape, which would affect the line cores, could also contribute to the CO2 profile deviations at higher altitudes.~~

Although the effect of typical perturbations in the instrument field of view, zero-level offset, and spectroscopic parameters is relatively small compared to the effect of temperature errors, the cumulative effect of these errors could explain the deviations from the truth in Fig. 9(a) and 10(a).

**Comment #5**
Figure 11: it might be instructive to add a further panel displaying XCO2 for each retrieval setup.

**Response:**
The difference in XCO$_2$ between scaling and profile retrievals is less than 5 ppm while XCO2 between the different days varies by ~20 pppm and putting XCO$_2$ from both profile and scaling retrieval on a same

panel in Fig. 11 wouldn't be clear. Figure R7 shows an updated Fig. 11 with the profile retrieval XCO2 and smoothed AirCore XCO2.

**Update:**

Changed text in **3.2.2 Information content and averaging kernel** to:

The RR is always smaller than 1 because the profile retrieval has more freedom to adjust the calculated spectrum and so can never produce larger residuals than scaling retrievals. Figure 12 also shows $XCO_2$ obtained from the scaling retrievals subtracted to $XCO_2$ obtained from profile retrievals for each window.

Added a panel to Fig. 11 in the preprint (Fig. 12 in the revised version)

[Figure]

**Figure 12: Shannon information content (top left), degrees of freedom for signal for the CO2 profile (top right), and ratio of residuals (bottom left), and profile minus scaling retrieval XCO2 (bottom right) for all Lamont spectra coincident within ±1 h of the AirCore last sampling time for AirCores launched on the dates indicated in the bottom righton the right. Each new date is marked by a vertical dashed line.**

**Comment #6**

Figure 13: Why is GGG XCO2 so significantly off from the in situ value (scaling retrievals in the TCCON bands are off by about 4 ppm)? I assume that required calibration factors have not yet been applied?

**Response:**

There is no airmass dependent correction factors or in-situ correction factors applied. This ~1% offset might be due to uncertainties in line intensity of ~1% in the TCCON windows. The retrieved VSFs in the TCCON windows in GGG2020 are typically ~1.01. Recent line intensity measurements at the US National

Institute of Standards and Technology (Long et al., 2020) in the region corresponding to the TCCON1 window can be made with ~0.1% uncertainty. A similar study is being conducted for the CO2 band in the TCCON2 window. We recently tested including these line intensity measurements in the linelist for the TCCON2 window, with these the retrieved VSF become ~1.002 instead of ~1.01 when using the AirCore as a priori.

**Comment #7**

Note that this problem of bias is connected to a significant problem in your retrieval setup: as far as I oversee, the CO2 a-priori profiles used are intended to match with the actual CO2 mixing ratios in the atmosphere. But the CO2 becomes calibrated to WMO scale only further down the processing line (by rationing CO2 column over the O2 column and application of ADCFs + AICFs). Let's assume the calibration is off: this poses no problem for a scaling retrieval (same residuals, same solution, only a different scaling factor is reported). However, an optimal estimation setup will misbehave if the a-priori is biased: it will try to approach the a-priori values in altitudes where the sensitivity is low and in effect the retrieved profile will oscillate significantly. I think it would be superior to apply a pure smoothness constraint as used by Dohe instead of an optimal estimation with diagonal constraint and just some correlation on top. If a pure smoothness constraint is used, a bias of the a-priori remains without effect even for a profile retrieval (same residuals, same solution, only different scaling factors on identical profile shapes). Moreover, this setup is preferable because it allows a smooth transition between the profile retrieval and the simple scaling retrieval (just increase the smoothing constraint further and further).

**Response:**

With the a priori covariance used in the study, a bias in the a priori $CO_2$ profile has little effect on the retrieved profile shapes until the bias becomes larger than ~3%. This would be very uncommon for $CO_2$ priors at high altitudes. All the AirCore profiles over Lamont are within the GGG2020 a priori ±1% for P < 0.9 atm. Fig. R9 shows the effect of a +3% and +10% offset to the a priori $CO_2$ profile. It shows the retrieved profiles stay close to the a priori at the highest altitudes where there is little information. And to match the measured spectrum this is compensated with less $CO_2$ retrieved at P > 0.8 atm in the Weak and TCCON windows.

To avoid issues when the a priori bias is unusually high, a scaling retrieval could be performed first, and the resulting profile could be used as a priori for the profile retrieval. But with the data used in the study the effect of a bias in the a priori is small compared to the effect of temperature errors, or compared to the effect of other types of errors that produce the oscillations observed in Fig. R3(a).

Figure R4 shows the difference in profiles retrieved with the AirCore as a priori, and with the AirCore increased by 3% as a priori. This is the difference between retrieved profiles shown in Fig. R3(a) and R3(c), but for all 10 Lamont days with AirCore profiles. Here it is clearer that the deviation at P > 0.8 atm is larger in windows that have less sensitivity to the lowest altitudes while the Strong window shows differences within ±3 ppm at P > 0.1 atm. And the 3% offset considered here is unrealistically high.

Comment 7 is also addressed in Response to comment 4. It would be interesting to evaluate different a priori constraints after the implementation of a temperature retrieval or correction.

[Figure]

**Figure R3: The left-hand panels show CO₂ profiles retrieved using real spectra on 14 January 2012. In (a), we use the AirCore profile as the a priori. In (c), the a priori is increased by 3%. In (e), the a priori is increased by +10%. The right-hand panels: (b), (d), and (f), show the difference between the retrieved profiles and AirCore, corresponding to (a), (c), and (e) respectively.**

[Figure]

**Figure R4: change in the profiles retrieved from real Lamont spectra when using AirCore as a priori and when using AirCore increased by 3% as a priori. Each window has 10 lines for each day with an AirCore profile over Lamont.**

**Response to RC1**

**Comment #1**

The two things that are missing are:

1) predicted errors propagated from the interferent errors, These errors are calculated empirically and should also be compared to their predicted impact, at least for temperature and water. These errors can be calculated using Jacobians, following Connor et al. (2016) Eq. 6, or Worden et al. (2004) (second term of Eq. 18).

**Response:**

This was done in Appendix D where we show the smoothing, noise, interference, and retrieval errors for retrievals with real spectra (Fig. D8). The interference error (which includes contribution from the water vapour profile scaling) is the smallest contribution, but it does not include the effect of temperature, which is not retrieved. GFIT does not compute the derivative of the spectrum wrt temperature. We would expect the interference error to increase after the implementation of a temperature retrieval.

We did not include estimates of the forward model error, which would require estimating a covariance matrix and computing the Jacobian for each of the non-retrieved forward model parameters (e.g. each spectroscopic parameter, temperature, surface pressure, etc.), that would correspond to $S_b$ and $K_b$ in

Connor et al. 2016. The effort of deriving $K_b$ is equivalent to that of implementing a temperature retrieval in GFIT2.

**Update:**

Figure D8 was updated to present results from the 8 days with Lamont data, with the mean+-standard deviation of each error components instead of just showing results from 2012-01-14. And the following was added at the end of Section 3.2.2 to bring more attention to this analysis:

Additional analysis of the vertical sensitivity of the retrieval is presented in Appendix D, as well as a decomposition of the retrieval error into the interference, measurement noise, and smoothing errors as shown in Fig. D8. The interference error is the smallest (<0.5%) contribution but does not include the effect of temperature errors. The measurement noise error decreases from ~1% at the surface to ~0.2% at pressures less than 0.6 atm (> 5 km), and the smoothing error dominates and decreases roughly from ~3% at the surface to 1% at the top of the atmosphere.

**Comment #2**

The performance of the profile retrieval with real spectra is only reported as a column result (e.g. Fig 13). The standard deviation and bias versus AirCore should be calculated at all pressures and summarized in the paper.

**Response:**

All the $CO_2$ profiles and difference profiles shown in Section 3 are for the average of retrieved profiles from spectra meeting the coincidence criterion with AirCore. The right hand panels of these figures show the differences with AirCore. Fig. 8 and 9 show results of retrievals from real spectra. However these are indeed missing the standard deviation, both from the set of profiles averaged, or as the square root of the diagonal elements of the retrieval covariance matrix. The latter is shown in Fig. D8, which was updated to show statistics over the 8 days with AirCore profiles over Lamont, instead of just on 2012-01-14. Although it is reported there as a fraction of the a priori uncertainty and may not be straightforward to interpret in parallel with the profile figures, especially since for real spectra the prior uncertainty is described by Equations 1-4 from section 2.2. A second panel was added to Fig. D8 with the a priori uncertainty profile itself. These errors were summarized as shown in the response to Comment #1.

The retrieval error ranges from ~40-100% of the a priori uncertainty, with that fraction roughly increasing with altitude, and the a priori uncertainty decreases with altitudes from ~5% (~20 ppm) at the surface to ~1% (~4 ppm) at 5 km altitude and above.

The differences between the retrieved $CO_2$ profiles and AIrcore profiles as shown in right panels of Fig. 8 and 9 are larger than the retrieval errors (obtained with the square root of the diagonal elements of the retrieval covariance matrix).

**Update:**

[Figure]

**Figure D8:** **The left panel shows the** square root of the diagonal elements of the retrieval total error covariance matrix $\hat{S}$, the null space covariance matrix $S_N$, the interference error covariance matrix $S_i$, and the measurement noise error covariance matrix $S_m$ expressed as a fraction of the a priori uncertainty $\sigma_a$. **Each line is the average from the set of 8 days with AirCore measurements over Lamont, and the bands indicate the standard deviation. The right panel shows the a priori uncertainty.**

**Comment #3**

Line 17. To help the reader understand the scope of the paper, I would change the second sentence of the abstract to, "With these improvements, CO2 profiles were obtained from sequential retrievals in five spectral windows with different vertical sensitivities using synthetic and real spectra."

**Response:**

The text was updated as suggested.

**Comment #4**

Add to the paper: It was unclear to me (or maybe I missed it) what is retrieved. Is surface pressure? Water profile? Water scaling? Temperature profile? Temperature scaling? This could be listed at the start of methods.

**Response:**

Table 2 presents all the retrieved parameters. In GFIT the state vector is only made of retrieved parameters.

**The caption of Table 2 was updated to:**

Components of the state vector in GFIT2 profile retrievals. These are all the retrieved parameters.

**Text update:**

Table 2 summarizes the components of the state vector used in GFIT2. Fifty-one VSFs are retrieved (one for each atmospheric level) for the primary target gas, while only one VSF is retrieved for each of the interfering species profiles (non-$^{12}C^{16}O_2$ species included in Fig. 1, except for "solar" and "other", other $CO_2$ isotopologues are only retrieved as interfering species in the Strong window).

**Comment #5**

Line 82. I would add, "... and would also allow TCCON to be used for validation of TIR satellite products, e.g. from AIRS and GOSAT/2, and vertically resolved NIR GOSAT and OCO-2 experimental products."

**Response:**

The text was updated as suggested.

In addition to the suggested text, a reference to AIRS was added on Line 41 where the other satellites are first mentioned.

**Added reference:**

Aumann, H. H., Chahine, M. T., Gautier, C., Goldberg, M. D., Kalnay, E., McMillin, L. M., Revercomb, H., Rosenkranz, P. W., Smith, W. L., Staelin, D. H., Strow, L. L. and Susskind, J.: AIRS/AMSU/HSB on the aqua mission: Design, science objectives, data products, and processing systems, IEEE Trans. Geosci. Remote Sens., 41(2 PART 1), 253–263, doi:10.1109/TGRS.2002.808356, 2003.

**Comment #6**

Line 117. I would add a sentence saying, "OCO-2/3 and GOSAT/2 use the Weak1 and Strong bands as well as a band centered at 0.765 um not used by TCCON."

**Response:**

Some TCCON stations with Si detectors measure the $O_2$ A-Band, although it is not part of public data releases and is indeed not used for deriving Xgas products. The OCO-2 window centered near 1.61 µm corresponds to the TCCON1 window rather than the Weak1 window.

**Update:**

OCO-2/3 and GOSAT/2 use two windows comparable to the TCCON1 and Strong windows to retrieve $CO_2$, and use the $O_2$ A-band (centered near 13158 cm$^{-1}$).

**Comment #7**

Line 147. The sentence, "We see no advantage to fitting noncontiguous windows in parallel, rather than in series, and then averaging the results." is incorrect. Mathematically these are very different. The way to fit sequentially is called a sequential update and discussed in Rodgers, 2000 or Dudhia et al., 2002 and involves setting the constraint for the next step to the error covariance from the previous step.

**Response:**

We recognize the wording used may be confusing. By "sequential retrieval" we mean that retrieved parameters are obtained from each spectral window separately, as opposed to a joint-band retrieval. This is different from the "sequential update" method described in Rodgers, 2000.

As described in Rodgers, 2000, the "sequential update" method does not present advantages in the n-form formulation, which is the one used in GFIT2. This should be clear from the description of the GFIT2 implementation presented in Appendix B. What Appendix B does not describe is the detail of the implementation of the "inversion step" behind equation B2.

GFIT2 uses the algorithm described in Rodgers 2000 "Which formulation for the linear algebra?: The n-form" (section 5.8.1.1) but for the Levenberg-Marquardt method and using the HFTI algorithm (Lawson and Hanson, 1974) for obtaining matrix pseudoinverses by solving problems of the type $Bx = I$, with $I$ the identity. The HFTI algorithm is also used to solve for the state update (equation B2 in the preprint).

Computational efficiency was not a concern for this study. By "advantage" we mean an improvement in the precision and accuracy of retrieved quantities, or a better vertical resolution. If the problem is linear enough, like is the case for $CO_2$, doing retrievals in each window separately and combining the results in post-processing should be equivalent to a joint-band retrieval but with the added possibility of diagnosing window-specific issues.

**Update:**

We see no advantage to fitting non-contiguous windows simultaneously , rather than separately , and then combining  the results. In TCCON post-processing the total columns retrieved from different retrieval windows ($CO_2$ from the TCCON1 and TCCON2 windows, for example) are averaged after removing window-dependent multiplicative biases, using retrieval errors as weights.

**Comment #8**

Line 189. It would help the reader to have the value for sigma listed for 100, 500, 200, and 1 hPa.

**Response:**

The a priori uncertainty is defined on an altitude grid and will thus vary slightly at given pressure levels. A figure of sigma with pressure was added in section 2.2, using the pressure grid of the a priori used for each day with Lamont spectra.

**Update:**

Added a figure and updated figure numbers throughout the text.

[Figure]

**Figure 3: A priori uncertainty profiles for each of the 8 dates presented in Table 3. These are defined by Eq. (1). Since σ is defined on an altitude grid, it varies slightly with pressure.**

**Comment #9**

Line 194. A 2 km length scale seems very narrow, particularly above 4 km. Just a comment.

**Response:**

The ensemble of ObsPack aircraft profiles over Lamont used to determine the length scale are typically between 0.5-5 km. The full width at half maximum of the rows of the correlation matrix built from these profiles was ~2 km, and we use the same value for the rest of the profile, which might indeed be inadequate above 5 km. Using a larger length scale above 5 km would increase the $CO_2$ profile deviations from the truth in profiles presented in section 3.2.1

**Comment #10**

Line 228. Add a sentence, "The DOFS are shown in figures 3-6, and 8-9."

Response:

The text was updated as suggested.

**Comment #11**

Figures 3-5. These errors should be compared to the predicted errors.

**Response:**

A paragraph was added before section 3.1.1

**Update:**

The total retrieval random error for the retrievals presented in this section is ~4.5% (~18 ppm), the contribution of random noise is ~0.8% (~3 ppm), see Appendix D for definitions of total and

measurement noise errors. When the deviations from the truth are larger than the a priori uncertainty (~20 ppm), it means the perturbation applied has a severe effect on the retrieval. Of course this can be mitigated by using a stronger a priori constraint or a measurement covariance matrix that reflects expected systematic errors, and not just random noise, but always at the cost of reduced sensitivity to CO2 too. The goal here is to estimate the relative effect of different kinds of expected systematic errors on retrieved profile shapes. Stronger constraints can only reduce the amplitude of the deviations from the truth, but the same structures would remain. When the perturbation to a parameter other than CO2 results in deviations from the truth much larger than those presented in Section 3.1.1, it means that errors in that parameter will dominate the variability in the retrieved CO2 profiles regardless of the retrieval constraints.

**Comment #12**

Figures 3-5. Question that might be out of the scope of this paper: do these errors affect TCCON XCO2? In other words can the same tests from Figures 3-5 be done for the standard TCCON retrieval?

**Response:**

Although the results were not shown in this study, each of the sensitivity tests were also performed with scaling retrievals. An upcoming paper by Laughner et al. following the GGG2020 public release will include details of the effect of different perturbations on new TCCON Xgas products.  Similar sensitivity tests have previously been presented for TCCON Xgas products in Wunch et al. 2011 and in the GGG2014 documentation (Wunch et al. 2015).

The effect of the sensitivity tests of Section 3.1 is shown in Figure R5. It shows the effect of each perturbation on the retrieved $XCO_2$ in the following way: $\Delta Bias$ is the average (for the 8 dates with AirCore profiles over Lamont) of the absolute change (relative to the unperturbed case) in the absolute difference between the retrieved and AirCore $XCO_2$. The labels of the horizontal axis of Fig. R2 indicate the perturbation applied. Thus, when a "Profile retrieval" bar is larger than a "Scaling retrieval" bar, it means the $XCO_2$ retrieved from the profile retrieval is more sensitive to the perturbation considered than the scaling retrieval. The perturbations are labelled A to L.

It shows that $XCO_2$ from profile retrievals is as expected less sensitive to the $CO_2$ a priori profile (perturbations B and C), but is also more sensitive to temperature errors (perturbations G-H-I) except in the Weak windows.  All windows are relatively insensitive to the $H_2O$ perturbation considered (perturbations D-E-F). Perturbation L shows the effect of using the Voigt line shape to fit a synthetic spectrum generated with the qSDV+LM line shape. There is no change in the Weak2 window as it does not have qSDV+LM line parameters implemented, and little change in the Weak1 window as qSDV+LM is only implemented for the interfering $CH_4$ lines in that window. Perturbation L illustrates how using the Voigt profile was the most important source of mismatch between retrieved and true $XCO_2$ in the TCCON windows.

[Figure]

**Figure R5: Absolute change in the absolute difference between the retrieved and true XCO₂ relative to the reference case (fitting a synthetic spectrum using the same a priori used to generate it), each bar is the average of that quantity from the 8 days with AirCore profiles considered in the study, and the error bars are the corresponding standard deviations. The labels of the horizontal axis indicate the kind of perturbation applied before fitting the synthetic spectrum.**

**Comment #13**

Figure 8. I do not understand how the result can be up to 50 ppm off when the a priori is set to 6 ppm

**Response:**

By "when the a priori is set to 6ppm" I assume "a priori uncertainty" was meant. The a priori uncertainty is ~5% (20 ppm) at the surface for the retrieval results of section 3.2. The response to **Comment #8** should also address this comment.

**Comment #14**

Table 5. My note: I have not found information content to be useful because it is not aware of many of the systematic errors in the system.

**Response:**

Like the averaging kernel, the DOFS give information on the retrieval sensitivity to $CO_2$ in the absence of systematic biases, this is useful information only on the potential of the profile retrieval.

**Comment #15**

Figure 13. My note: The variability of the results for xco2 for the different bands was surprising.

**Response:**

The variability between different lines is caused by window specific biases. And the error bars of ~4-5 ppm are that large, even for the scaling retrievals, because the retrieval errors were not scaled like is done in TCCON post-processing. This scaling of the retrieved error is done to better reflect the variability of $XCO_2$ in consecutive spectra, since the retrieval error is an overestimate of the random error due to the presence of systematic residuals. The scaling is illustrated in Fig. R6 with a set of GGG2014 TCCON data from Eureka in 2018.

[Figure]

**Figure R6: $XCO_2$ error derived from the retrieval error (orange), and after a scaling is applied to better reflect the $XCO_2$ variability from consecutive spectra (blue), example from Eureka 2018 GGG2014 TCCON data.**

**Response to RC2**

**Comment #1**

Line 26-28: please add 'at Lamont' at the end of the sentence

**Response:**

We believe that the high sensitivity of the profile retrieval to temperature errors is not specific to the Lamont site, but to the retrieval algorithm, so we left the sentence as is.

**Comment #2**

Line 61: "non-linear least-squares spectral fitting algorithm" is GFIT also based on the OEM? If yes, please mention it in the text. If not, please add some discussion about the retrieval method in the paper.

**Response:**

Yes GFIT is based on the optimal estimation method, specifically GFIT implements the "(n+m)×n" problem described in Rodgers, 2000 "Numerical Efficiency. Which formulation for the linear algebra? The n-form".

**Update**:

 Technically, GFIT handles the scaling retrieval by weakly constraining the fitted VSF factor. The approach is equivalent to performing an optimal estimation of the VSF, assigning a value of unity to the a-priori VSF and a value of 106 as its expected range of variability.

**Comment #3**

Line 62: "A forward model . . . a priori knowledge of atmospheric conditions", please also add instrumental conditions and the observation geometry.

**Update:**

"A forward model computes an atmospheric transmittance spectrum for a given observation geometry using a priori knowledge of atmospheric conditions and assuming a perfectly aligned instrument."

**Comment #4**

Line 66: "Scaling retrievals do not require inter-level constraints on a priori concentration uncertainties. ". A scaling retrieval is actually equal to a very strong inter-level constraint. Please re-write this sentence.

**Response:**

The scaling retrieval is indeed equivalent to a profile retrieval with very strong interlayer constraints, but this constraint is not adjustable in a scaling retrieval. The text was updated as suggested in the Editor's comment and as shown in the response to Comment #2.

**Comment #5**

Line 67. "an uncertainty of $10^6$ ". Please check why the uncertainty is so large?

**Response:**

The a priori uncertainty is so large to allow unbiased retrievals in situations where we have no a priori knowledge of the gas amount. There is no advantage of imposing a tighter constraint because a scaling retrieval is much better conditioned than the profile retrieval and the a priori total column does not need

to be close to the true column for a successful retrieval. However, the profile shape is implicitly infinitely constrained and cannot change from the a priori profile shape.

**Comment #6**

Line 101. Does the wavenumber scales not included in GGG2014? If not, please mention it in the text.

**Update:**

(2) a solar- gas stretch fitted to account for Doppler-driven differences between solar and telluric wavenumber scales, in GGG2014 only the stretch in the telluric wavenumber scale was fitted

**Comment #7**

Line 107: change "original TCCON spectral windows " to "original TCCON retrieval windows"

**Response:**

The text was updated as suggested.

**Comment #8**

Line 148:" We see no advantage to fitting non- contiguous windows in parallel, rather than in series, and then averaging the results. ". How do you average the results? Do you apply the weighting function based on the SNR of each window?

**Response:**

In TCCON post-processing the total columns retrieved from different retrieval windows ($CO_2$ from the TCCON1 and TCCON2 windows, for example) are averaged after removing window-dependent multiplicative biases, using retrieval errors as weights.

A different method was tested to combine the profile retrieval results but was not shown and still requires more work. But the principle is to obtain the combined profile $x_r$ from the profiles retrieved in N windows as:

$$x_r = S_r \left( \left[ \sum_{i=1}^{N} \hat{S}_i^{-1} \hat{x}_i \right] - (N-1) S_a^{-1} x_a \right)$$

$$S_r = \left( \sum_{i=1}^{N} S_a^{-1} + K_i^T S_{y,i}^{-1} K_i \right) - (N-1) S_a^{-1}$$

$$A_r = \left( S_a^{-1} + \sum_{i=1}^{N} K_i^T S_{y,i} K_i \right)^{-1} \sum_{i=1}^{N} K_i^T S_{y,i}^{-1} K_i$$

where $(N-1)$ contributions of the a priori are subtracted to obtain expressions equivalent to a joint-band retrieval. These combined profiles tend to be strongly weighted towards the profile retrieved from the Strong window and can amplify deviations from the truth caused by systematic errors. The removal of window-dependent biases as done for total columns in the TCCON post-processing was not straightforwardly applicable to retrieved profiles and will be the subject of future work.

**Update:**

We see no advantage to fitting non-contiguous windows smultaneously , rather than separately , and then combining  the results. In TCCON post-processing the total columns retrieved from different retrieval windows ($CO_2$ from the TCCON1 and TCCON2 windows, for example) are averaged after removing window-dependent multiplicative biases, using retrieval errors as weights.

**Comment #9**
Line 312-314:As the GFIT2 does not allow retrieve 2 profiles simultaneously, have you ever tried to retrieve the $H_2O$ profile first, and use the retrieved $H_2O$ profile as the a priori profile to do the $CO_2$ retrieval?

**Response:**

No, we did not try this. This would be interesting to investigate after the implementation of a temperature retrieval.

**Comment #10**
Line 444: What is the physical reason for the 'zero-level offset' error?

**Response:**

Zero-level offsets can be caused by detector non-linearity or aliased out-of-band spectral signal.

**Comment #11**
Line 445: What do you mean by "higher altitudes"? please note the vertical range specifically

**Response:**

>10 km was meant.

Following the Editor's comment, additional sensitivity tests were added as an Appendix with perturbations to the instrument field of view (which affects the ILS width), and to ZLO. And the text of Line 443-446 was changed.

**Update:**

~~The cause of the deviations in Fig. 8(a) and 9(a) could be due to errors in the zero-level offset in the Weak and TCCON windows, where it is assumed to 445 be zero. In these windows the zero-level offset is not fitted as they lack saturated absorption lines. Errors in the instrument line shape, which would affect the line cores, could also contribute to the CO2 profile deviations at higher altitudes.~~

Although the effect of typical perturbations in the instrument field of view, zero-level offset, and spectroscopic parameters is relatively small compared to the effect of temperature errors, the cumulative effect of these errors could explain the deviations from the truth in Fig. 9(a) and 10(a).

**Comment #12**
Line 545. Section 3.2.4, I expect the authors to present a table here summarizing all the uncertainties for the $CO_2$ profile at Lamont, together with the typical vertical variation of $CO_2$.

**Response:**

The profiles of uncertainty like Figure D8 should be more informative than a table for the uncertainty on retrieved profiles. Figure D8 shows profiles of noise, smoothing, interference, and retrieval errors which are representative of all the $CO_2$ profiles at Lamont. The last sentence of section 3.2.2 was updated to draw more attention to and summarize this analysis.

**Update in 3.2.2:**

Additional analysis of the vertical sensitivity of the retrieval is presented in Appendix D, as well as a decomposition of the retrieval error into the interference, measurement noise, and smoothing errors as shown in Fig. D8. The interference error is the smallest (<0.5%) contribution but does not include the effect of temperature errors. The measurement noise error decreases from ~1% at the surface to ~0.2% at pressures less than 0.6 atm (> 5 km), and the smoothing error dominates and decreases roughly from ~3% at the surface to 1% at the top of the atmosphere.

**Update in 3.2.4:**

Even when the errors due to the a priori meteorology are minimized, deviations from the truth due to instrument misalignment, radiative transfer, sun-tracker pointing, or uncertainties in line parameters are larger than  the steepest vertical CO2 gradients (~5 ppm/km) observed in the ensemble of aircraft profiles from NOAA's ObsPack.

**Updated figure D8 (see Response to RC1 Comment #2).**

**References**

Lawson, L., and Hanson, R. J., Solving Least Squares Problems, Prentice-Hall, Inc., 1974, Chapter 14.

Long, D. A., Reed, Z. D., Fleisher, A. J., Mendonca, J., Roche, S. and Hodges, J. T.: High-Accuracy Near-Infrared Carbon Dioxide Intensity Measurements to Support Remote Sensing, Geophys. Res. Lett., 47(5), doi:10.1029/2019GL086344, 2020.

Rodgers, C. D.: Inverse Methods for Atmospheric Sounding: Theory and Practice, World Scientific, Singapore, 2000.

Wunch, D., Toon, G. C., Blavier, J.-F. L., Washenfelder, R. A., Notholt, J., Connor, B. J., Griffith, D. W. T., Sherlock, V. and Wennberg, P. O.: The Total Carbon Column Observing Network, Philos. Trans. R. Soc. A Math. Phys. Eng. Sci., 369(1943), 2087–2112, doi:10.1098/rsta.2010.0240, 2011b.

Wunch, D., Toon, G. C., Sherlock, V., Deutscher, N. M., Liu, C., Feist, D. G. and Wennberg, P. O.: Documentation for the 2014 TCCON Data Release (Version GGG2014.R0). CaltechDATA, doi:https://doi.org/10.14291/tccon.ggg2014.documentation.r0/1221662, 2015.